# Multimodal in vivo recording using transparent graphene microelectrodes illuminates spatiotemporal seizure dynamics at the microscale

Nicolette Driscoll [1,2,3,14], Richard E. Rosch [1,4,5,14], Brendan B. Murphy[1,2,3], Arian Ashourvan [1,2], Ramya Vishnubhotla[6], Olivia O. Dickens [7], A. T. Charlie Johnson [6], Kathryn A. Davis[2,8], Brian Litt[1,2,8], Danielle S. Bassett [1,6,8,9,10,11], Hajime Takano [8,12✉] & Flavia Vitale [2,3,8,13✉]

Neurological disorders such as epilepsy arise from disrupted brain networks. Our capacity to treat these disorders is limited by our inability to map these networks at sufficient temporal and spatial scales to target interventions. Current best techniques either sample broad areas at low temporal resolution (e.g. calcium imaging) or record from discrete regions at high temporal resolution (e.g. electrophysiology). This limitation hampers our ability to understand and intervene in aberrations of network dynamics. Here we present a technique to map the onset and spatiotemporal spread of acute epileptic seizures in vivo by simultaneously recording high bandwidth microelectrocorticography and calcium fluorescence using transparent graphene microelectrode arrays. We integrate dynamic data features from both modalities using non-negative matrix factorization to identify sequential spatiotemporal patterns of seizure onset and evolution, revealing how the temporal progression of ictal electrophysiology is linked to the spatial evolution of the recruited seizure core. This integrated analysis of multimodal data reveals otherwise hidden state transitions in the spatial and temporal progression of acute seizures. The techniques demonstrated here may enable future targeted therapeutic interventions and novel spatially embedded models of local circuit dynamics during seizure onset and evolution.

---

[1] Department of Bioengineering, University of Pennsylvania, Philadelphia, PA, USA. [2] Center for Neuroengineering and Therapeutics, University of Pennsylvania, Philadelphia, PA, USA. [3] Center for Neurotrauma, Neurodegeneration, and Restoration, Corporal Michael J. Crescenz Veterans Affairs Medical Center, Philadelphia, PA, USA. [4] MRC Centre for Neurodevelopmental Disorders, King's College London, London, UK. [5] Department of Paediatric Neurology, Great Ormond Street Hospital for Children NHS Foundation Trust, London, UK. [6] Department of Physics and Astronomy, University of Pennsylvania, Philadelphia, PA, USA. [7] Graduate Group in Biochemistry and Molecular Biophysics, University of Pennsylvania, Philadelphia, PA, USA. [8] Department of Neurology, Perelman School of Medicine, University of Pennsylvania, Philadelphia, PA, USA. [9] Department of Psychiatry, Perelman School of Medicine, University of Pennsylvania, Philadelphia, PA, USA. [10] Department of Electrical & Systems Engineering, University of Pennsylvania, Philadelphia, PA, USA. [11] Santa Fe Institute, Santa Fe, NM, USA. [12] Division of Neurology, Children's Hospital of Philadelphia, Philadelphia, PA, USA. [13] Department of Physical Medicine and Rehabilitation, University of Pennsylvania, Philadelphia, PA, USA. [14]These authors contributed equally: Nicolette Driscoll, Richard Rosch. ✉email: takanoh@email.chop.edu; vitalef@pennmedicine.upenn.edu

Understanding the dynamics underlying seizure generation and spread at the scale of neural microcircuits is critical for improving epilepsy therapies, and enabling more efficient seizure control. For example, targeted epilepsy treatments such as surgical resection, thermal ablation, or responsive neurostimulation rely on precisely localizing seizure onset regions to be effective. Currently, clinicians localize these regions by observing seizure semiology, identifying electrophysiological signatures from millimeter-scale implanted electrodes, and combining these features with noninvasive imaging techniques, such as magnetic resonance imaging and positron emission tomography. Unfortunately, the spatial and temporal resolution of these techniques is limited to collective, region-wide network oscillations, which hampers the development of precision therapy approaches. A growing body of literature indicates that seizures, in fact, begin at the scale of neural microcircuits, which current clinical technologies cannot observe[1–4]. Thus, there is great interest in observing cellular and microcircuit level dynamics as seizures begin and spread, and in linking these to the macroscale, brain-wide activity recorded by current clinical technologies.

Recently, microelectrode array recordings in humans have opened a window into complex, microscale ictal dynamics within the cortical area usually covered by a single clinical electrocorticography (ECoG) electrode. Similarly, microarray and microwire studies have revealed spatially distinct territories near the seizure onset zone[5], characterized by heterogeneous firing of single cells[6] and associated with different states of excitation–inhibition balance[7]. Observing seizures at this scale has led to the idea that the cortical areas involved can be divided into a core territory of recruited neurons (the "ictal core"), surrounded by areas that receive intense synaptic input, but do not show hypersynchrony and pathologically high firing rates (the "ictal penumbra"). Observations at the microscale have also revealed previously unobserved temporal phenomena at play during epileptic seizures: individual ictal and interictal epileptiform discharges observed on intracranial EEG (iEEG) can spread near-synchronously across macroscopic brain areas[8,9]. In contrast, microscale neurophysiological markers of the so-called "ictal wavefront" spread locally at much slower speeds, typically not exceeding ~5 mm/s (ref. [10]). It has now become apparent that macroscale signatures of these microscale dynamics are present in particular frequency bands of clinical iEEG, but were only identified with the benefit of concurrent microelectrode recordings[11,12].

Although microelectrode array recordings are advancing our understanding of microscale epileptic dynamics, these electrode arrays still cannot offer a complete picture of neural circuits due to their limited spatial sampling. Recordings with the Utah array (96 electrodes in a 4 × 4 mm area), for example, typically detect the activity from several dozen up to ~180 neurons, at most, from patches of cortex that are just a few millimeters across, and which contain hundreds of thousands of neurons[6]. Evolving alongside implantable microelectrode technologies, optical tools and techniques enable investigators to observe neural activity at cellular and even subcellular spatial scales[13–15]. Optogenetics gives these techniques tremendous power to image and modulate the function of specific, genetically targeted cell types in the brain to discern the roles of different cell assemblies[16]. Calcium fluorescence imaging has provided insights into the role of inhibitory interneurons in seizure initiation and spread[4], and in the microstructure of synchronous neuronal assemblies in epileptic networks[17–19]. However, such imaging studies typically suffer from low temporal resolution limited by the kinetics of the fluorescent reporter molecules, as well as the speed of currently available imaging systems[14,20,21].

Combining electrophysiological and optical recording modalities presents a unique opportunity to harness the spatial resolution of optical imaging along with the temporal resolution of electrophysiology. However, such a combination requires imaging and recording simultaneously in the same location, which is not possible with conventional microelectrode arrays composed of opaque metallic materials, as these block optical access and suffer from photoelectric artifacts[22]. Recently, progress in transparent microelectrode array technology, predominantly utilizing graphene, has enabled such multimodal studies[23–26]. However, innovation is still needed in experimental and analytical methods to fully realize the potential of these multimodal mapping techniques. Notably, there is currently a lack of analytical methods to combine high-density optical and neurophysiology datasets, while accounting for their differences in temporal and spatial sampling.

In this work, we demonstrate an experimental and methodological paradigm, which combines mapping of microscale local network dynamics at high spatiotemporal resolution with a quantitative analysis framework to distill the dynamics underlying seizure generation and evolution in vivo. Specifically, we utilize custom-made, transparent graphene microelectrode arrays to simultaneously record electrophysiology and optical signals of microscale brain activity in a rodent model of acutely induced epileptic seizures. We combine dynamic data features from both modalities in a sparsity-constrained nonnegative matrix factorization (NMF) approach, in order to identify spatiotemporal patterns across these acquisition modalities. NMF is used for dimensionality reduction, similar to principal component analysis, but yields sparser and potentially more interpretable factors due to the nonnegativity constraints on both the original feature matrix and the resultant factors. Such an approach has previously been used to decompose dynamic functional connectivity into sets of variably expressed subnetworks and track their temporal evolution in human neuroimaging[27–29]. This analysis reveals how the temporal progression of ictal electrophysiology is linked to the spatial evolution of the recruited seizure core. We show that locally synchronous increases in broadband micro-ECoG (μECoG) power are associated with a progressive expansion of the ictal core at seizure onset, and that state transitions occur during an established seizure that are accompanied by subtle changes in ictal discharge morphology. These microscale spatiotemporal dynamics are not apparent from electrophysiology alone, and highlight the unique advantages of multimodal recording enabled by transparent electronics. The analytical framework presented here serves as a proof of concept for future studies combining data from multiple modalities, with the potential to illuminate mechanisms underlying seizure initiation, spread, and microscale network interactions that may characterize other types of brain disorders.

## Results

**Optimized fabrication produces optically transparent and low-impedance graphene microelectrode arrays.** To enable multimodal brain recording, we designed and fabricated transparent and flexible graphene microelectrode arrays using a parylene-C substrate, with electrodes and traces within the array consisting of two stacked layers of graphene grown via chemical vapor deposition and doped with nitric acid ($HNO_3$). In each device, graphene forms the electrodes and conductive traces in the distal end of the array, and overlaps with Ti/Au conductive traces outside of the array, such that the entire recording area of the device is fully transparent (Fig. 1a). The electrode array consists of a 4 × 4 grid of 50 μm × 50 μm square electrode contacts with 500 μm pitch (total recording area of 1.55 mm × 1.55 mm and

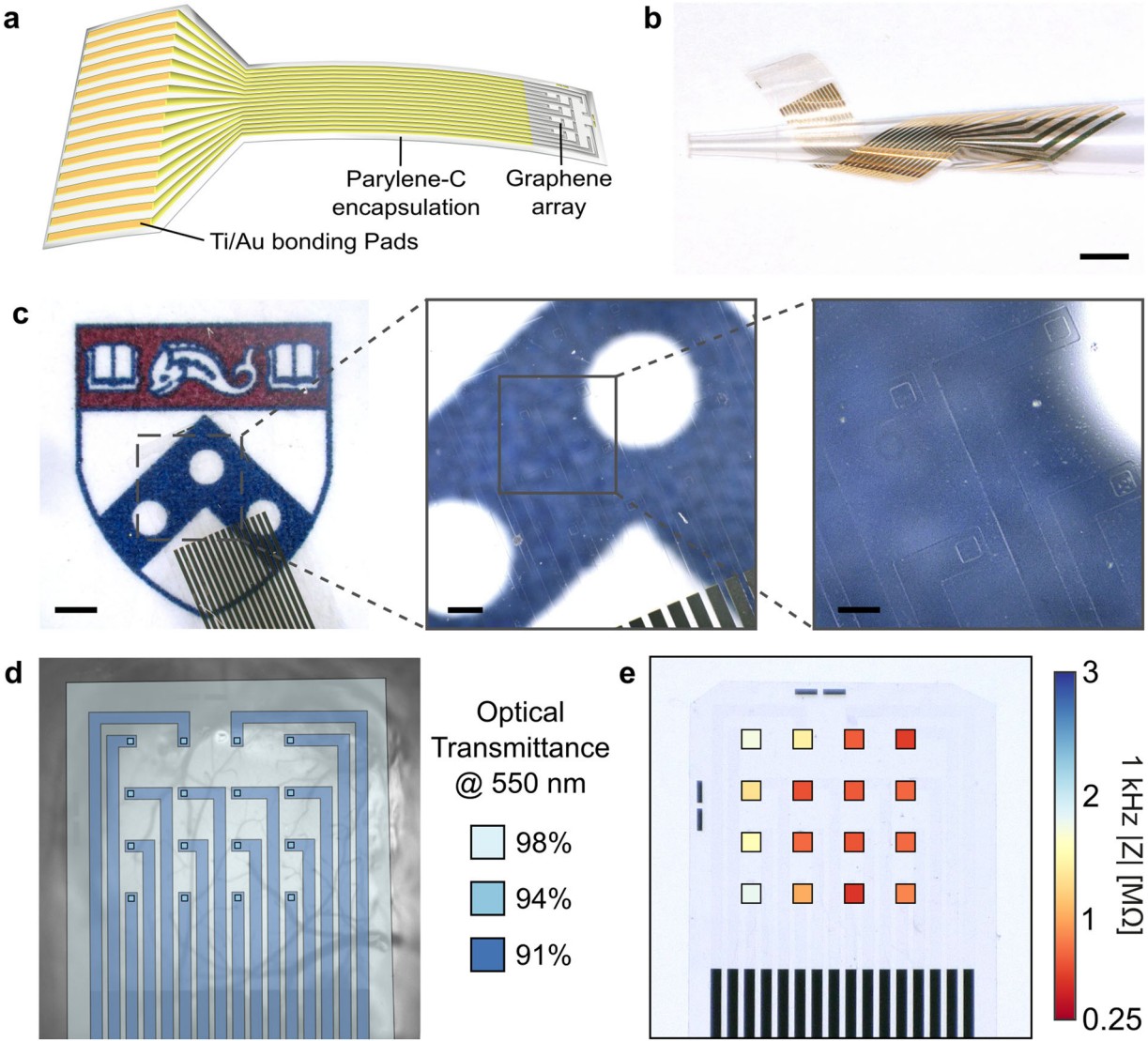

**Fig. 1 Optically transparent graphene microelectrocorticography arrays. a** Schematic of graphene µECoG device: a fully transparent array of graphene microelectrodes and traces connects to gold traces in a flexible cable, which can be interfaced with a data acquisition system through a ZIF connector. **b** A graphene µECoG device wrapped around a pipette tip to demonstrate its flexibility. Scale bar: 2 mm. **c** Images of the graphene electrode array overlaid on the Penn logo to demonstrate optical transparency. Scale bars, left to right: 1 mm, 250 µm, and 100 µm. **d** Optical transmittance at 550 nm for different locations in the graphene µECoG array. Overall, the device is >90% transparent across the visible to near infrared spectrum. **e** 1 kHz impedance magnitude for each 50 µm × 50 µm transparent graphene electrode in the array, averaged over three devices. Values are overlaid in the corresponding electrode locations on a light microscopy image of a single device. For clarity, electrodes are not shown to scale.

total array footprint 2.75 mm × 2.75 mm) patterned on a 4-µm thick parylene-C substrate and encapsulated by another 4-µm thick parylene-C layer. The combination of the thin profile of the device, the low Young's modulus of parylene-C and the extreme broadband optical transmittance of both parylene-C and graphene endow the device with high flexibility (Fig. 1b) and optical transparency (Fig. 1c). A schematic of the electrode fabrication process is shown in Supplementary Fig. S1, and a detailed description of the fabrication process can be found in the "Methods" section. To determine processing conditions which would minimize electrode impedance, we characterized the effects of stacking multiple graphene layers, and utilizing $HNO_3$ chemical doping on the electrode impedance and charge storage capacity (see Supplementary Methods and Supplementary Fig. S2). We determined that devices fabricated with two-layer graphene which were $HNO_3$-doped in a layer-by-layer fashion achieved >90% optical transparency across the visible spectrum

(Fig. 1d and Supplementary Fig. S3). The average impedance over four such devices ($n = 64$ channels) was $908 \pm 488$ kΩ at the 1 kHz reference frequency (Fig. 1e and Supplementary Fig. S2), which corresponds to an area-normalized impedance of $22.7 \pm 12.2$ Ω·cm$^2$.

**Transparent graphene electrodes enable multimodal in vivo recording of epileptic seizures.** We utilized our transparent graphene arrays to perform simultaneous calcium epifluorescence imaging of neurons and electrophysiology recording in vivo in an acute murine model of epilepsy. GCaMP6-expressing mice were anesthetized, and the graphene µECoG arrays were placed on the cortical surface following craniotomy. Electrode arrays were connected to a NeuroNexus amplifier/digitizer through a custom interface connector for electrophysiological recording, and widefield epifluorescence microscopy was used to simultaneously

capture GCaMP6 activity at low magnification at a frame acquisition rate of 10 Hz. Several previous studies combining electrophysiology and neuroimaging with transparent micro-electrodes have utilized two-photon imaging[25,26], which offers high spatial resolution but a limited field of view, thus providing detailed information for only a small population of cells lying beneath one or a small number of individual electrodes. For this study, we instead used widefield epifluorescence imaging to observe the calcium dynamics across the entire millimeter-scale cortical region covered by the array in order to maximally leverage data features from each modality. This choice also allowed us to observe the spatial dynamics of seizure spread across a millimeter-scale cortical area. The flexible electrode arrays readily conformed to the cortical surface and their high broadband optical transparency allowed penetration of the excitation light (470 nm) and fluorescence emission light (500–550 nm). A potassium channel blocker, 4-aminopyridine (4-AP), was bath-applied to induce epileptiform activity. We provide a schematic of the recording and imaging setup (Fig. 2a), and an image of the graphene electrode array on the cortex, outlined in white to show the locations of the transparent electrodes (Fig. 2b). Seizure events were recorded on the graphene array and were visible in epifluorescence imaging through the electrodes as changes in normalized fluorescence, where each imaging pixel was normalized by the amplitude and standard deviation of fluorescence changes observed in the baseline, pre-seizure condition.

The advantages of our experimental setup are underscored in Fig. 2. With the high-density graphene μECoG arrays, we obtained high temporal resolution recordings of seizure activity, including the fast dynamics of seizure onset as well as discrete high-frequency oscillations (HFOs). HFOs are of particular interest because of their potential role in ictogenesis and their use as a biomarker to identify seizure onset regions[30–32]. Throughout our study, we observed 200–300 Hz "fast-ripple" HFOs consistent with a pathologic, seizure-prone state as determined by a clinician trained in EEG marking (Fig. 2c). These oscillations are too brief and high frequency to be observed with calcium imaging alone. It is also notable that our transparent graphene electrodes recorded seizure activity with high signal-to-noise ratio (SNR; Supplementary Fig. S4). Despite some variability in electrode impedance across the array during in vivo recordings, we found that all electrodes had SNR > 5, and that SNR was not in fact correlated with electrode impedance (Supplementary Fig. S4a, b). Instead, the SNR values across the electrode array mapped more closely to the seizure onset pattern, with electrodes closer to the seizure focus showing larger amplitude ictal spikes and thus higher SNR (Supplementary Fig. S4c). While higher impedance electrodes do pick up more 60 Hz noise interference, this artifact is removed in the data filtering step such that electrode impedance does not affect the results of our analysis.

While electrophysiological recordings offer the high temporal resolution necessary to observe fast activity, such as pathologic HFOs, the spatial resolution of these recordings is limited by the minimum size, and spacing achievable for microelectrodes in an array. Despite recent advances in the development of ultrahigh-density microelectrode arrays[33], imaging modalities still offer superior spatial resolution for observing the spatial spreading dynamics of neural activity, albeit with limited temporal resolution. Using calcium fluorescence imaging, we observed complex, non-stereotyped spatial patterns of seizure initiation and spread, which we illustrate by plotting the putative ictal wavefront evolution over time for two different seizure events (Fig. 2d).

### Calcium imaging shows slow spatial evolution of the ictal core.
We show multimodal data recorded during one acute seizure onset in Fig. 3 and Supplementary Video S1. The locations of the

graphene electrodes on the cortex, and their position relative to the 4-AP application site, are shown in Fig. 3a. Upon visual inspection of the ECoG time series, we determined the time of ictal onset by identifying changes in rhythmic background activity that evolve in time and space; the determination was validated by a board certified epileptologist (KAD). In the recording event shown here, μECoG traces first show an increasing frequency of epileptiform discharges which appear near-synchronously across the whole electrode array, followed by a higher frequency, evolving rhythm that changes in amplitude and frequency for the duration of the seizure (Fig. 3b).

Associated with these changes in the electrophysiological activity, we observe an increase in high-amplitude calcium transients, followed by a transition of the global calcium fluorescence to a higher overall amplitude. When fluorescence is instead measured at the location of individual μECoG contacts, we further see a separation in the temporal dynamics of this progression of seizure-associated calcium activity. While individual calcium transients beneath each electrode appear to be timed nearly synchronously, channels to the left of the array (nearer the location of chemoconvulsant drug application) show an earlier and higher amplitude transition into the seizure state overall. This spatial evolution of the seizure is apparent in the sequence of normalized calcium images (Fig. 3c), where high-amplitude calcium fluorescence indicates that ictal activity evolves slowly from the bottom left corner of the image to reach maximum amplitude near the end of the recorded segment shown here. High-amplitude fluorescence corresponds to temporal summation of very fast neuronal firing, indicative of recruitment into the seizure. This intense, hypersynchronous activity in the recruited "ictal core" stands in stark contrast to the less synchronous and lower-frequency firing behavior of surrounding cortical areas, commonly referred to as the "ictal penumbra". The distinct neuronal dynamics of these areas are separated by a slowly traveling "ictal wavefront" that gradually expands throughout the recording to ultimately engulf nearly the entire field of view (Fig. 3c, d). Features of this core-penumbra organization of epileptic activity have previously been described both in human microelectrode recordings in chronic epilepsy, and in vitro using optical imaging of acutely induced seizures in cortical slices, where these dynamics unfold across temporal and spatial scales comparable to the ones reported here[7]. Similar to previous reports[7,34], individual ictal discharges may spread near-synchronously across the whole μECoG recording array, while calcium fluorescence, and to some extent signal amplitude on the μECoG array, expands slowly across the recorded surface. We treat this slow spatial evolution of the seizure as a traveling wave phenomenon[9], where we consider the boundary between significantly increased calcium fluorescence and activity nearer the resting state background level to be the edge of the ictal core (Fig. 3d).

Despite the apparently synchronous onset of ictal features across individual channels in the μECoG, there is a clear relationship between the spatial distribution of the calcium signal, and local features of the electrophysiological recording. Specifically, we observe a correlation between normalized μECoG broadband power (Pearson's correlation coefficient $r = 0.71$, $p < 0.001$ for linear fit), and local average normalized calcium fluorescence (Fig. 3e). While less obvious in the μECoG signal alone, this observation indicates that both calcium imaging and electrophysiology capture related, local spatiotemporal phenomena of ictal onset.

### Multimodal imaging identifies sequences of state transitions during seizure onset and evolution.
We next aim to describe the spatiotemporal profile of ictal onset across multimodal data in a

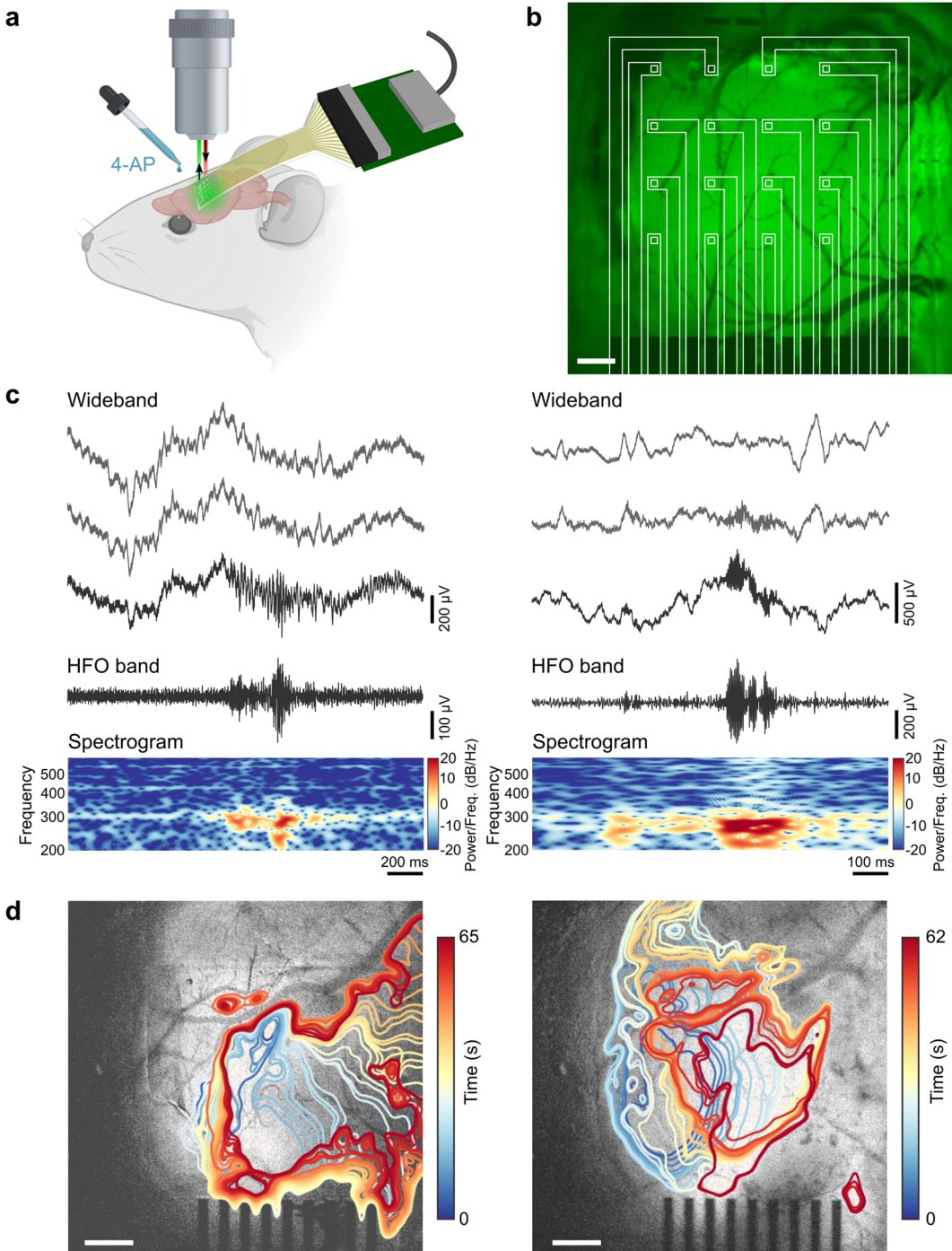

**Fig. 2 Simultaneous electrophysiology and calcium imaging of 4-AP-induced seizure activity in an anesthetized mouse. a** Schematic of the in vivo recording and imaging setup. Image created with BioRender.com. **b** Image of the graphene electrode array placed over barrel cortex, with baseline calcium epifluorescence visible in green, and array outlined in white to show locations of the electrodes. Scale bar: 300 μm. **c** Examples of two clinician-validated high-frequency oscillations (HFOs) recorded on graphene electrodes. Raw, wideband signal from three adjacent electrodes highlights the localized nature of the HFOs. Below these, the 80–600 Hz bandpass-filtered signal and the spectrogram, which reveals 200–300 Hz power consistent with fast-ripple HFOs, are shown for the bottom-most wideband signal trace. **d** Two representative examples of ictal wavefront onset and propagation patterns observed in the calcium epifluorescence imaging for different seizure events, showing complex non-stereotyped spatial patterns of activation and spread. Ictal wavefront progression is overlaid on images of baseline fluorescence. Scale bars: 500 μm.

lower-dimensional representation. We make the assumption that the high-dimensional dynamics of seizure onset and early ictal evolution can be well captured through time-varying expression of latent spatial factors and subnetworks, whose individual makeup does not change with time, but fluctuates in its relative expression. First, we calculate features previously identified as

dynamic markers of ictal territories[5] using temporal windows matching the calcium imaging frame acquisition speed: phase-locked high gamma, high gamma power, and local network synchrony. This choice enables us to downsample the μECoG data, while preserving the information afforded by its high temporal resolution, and align in time with the calcium fluorescence

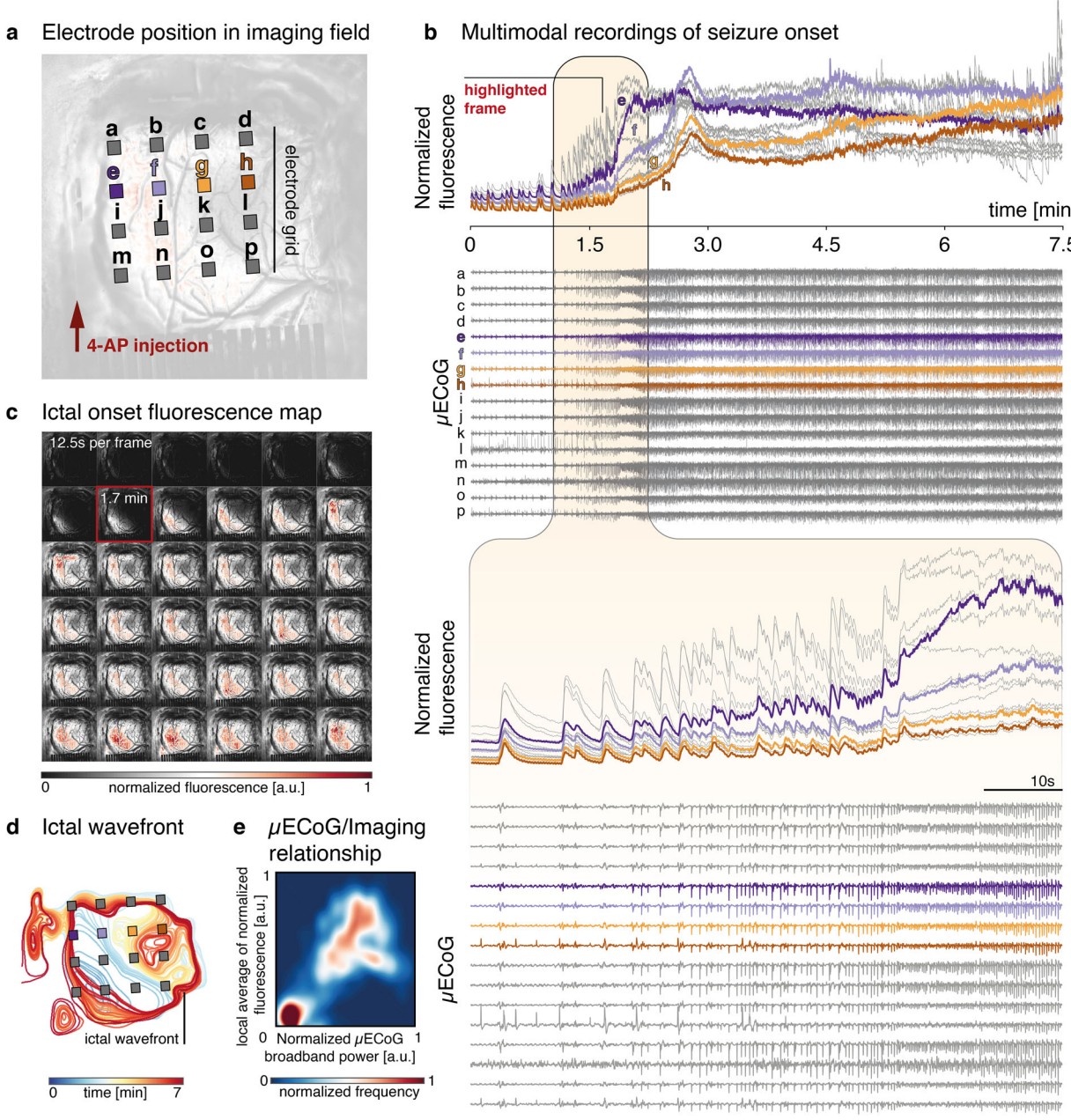

**Fig. 3 Concurrent multimodal data obtained during a single seizure onset. a** Relative positions of the graphene contacts on the cortex (individual electrodes not shown to scale) and the location of 4-AP drug application. Electrodes labeled e, f, g, and h are highlighted and shown as example data. **b** Fluorescence intensity (normalized to baseline mean and variance) beneath each individual electrode and simultaneous μECoG recordings from the graphene array. The period of acute seizure onset is magnified in the bottom panel. **c** Series of image frames corresponding to the seizure onset shown in **b**. From the full image sampling resolution of 10 Hz, frames are sampled here for visualization only every 12.5 s (0.08 Hz), illustrating the entire progression of seizure onset and propagation. **d** Ictal wavefront evolution over time for the seizure shown in **b**. Note particularly the early recruitment of channel e and the comparatively late recruitment of channel h. **e** Density plot indicating the significant relationship (Pearson's correlation coefficient $r = 0.71$, $p < 0.001$ for the linear fit) between μECoG broadband power and local calcium fluorescence averages. The colormap indicates the density, or normalized observed frequency of values distributed along the two dimensions (μECoG broadband power, calcium fluorescence).

images. To detect latent spatial factors and subnetworks that best explain the temporal variation in key data features, we use a sparsity-constrained NMF[35,36] with the number of factors chosen to optimize the variance explained per factor for a limited total number of factors used for dimensionality reduction (Supplementary Fig. S5). This method yields interpretable and sparse individual factors, while remaining computationally tractable (Fig. 4a). With this approach, we are able to integrate insights across different recording modalities, and to include many data

features to trace latent dynamics that might not otherwise be easily discernible from single data features alone[37].

The spatial evolution of the ictal core is apparent both in the calcium imaging- and electrophysiology-derived components of the factors obtained with NMF. For example, as we move from factor 1 to factor 6, we observe an increase in both μECoG broadband power and calcium fluorescence amplitude (Fig. 4b), which at the time of seizure onset is maximal at the bottom left corner of the array, closest to the drug application site. Similar to

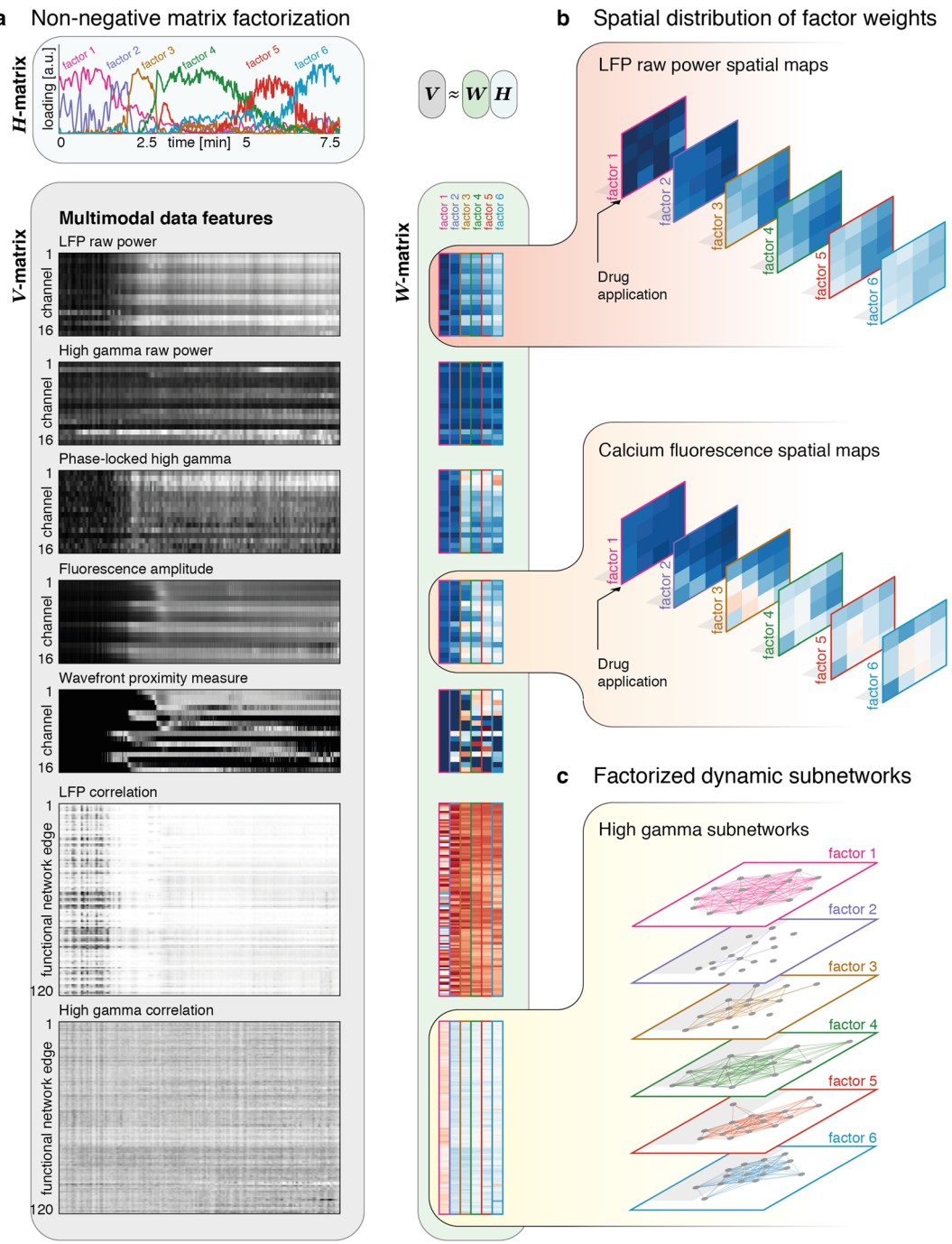

**Fig. 4 Nonnegative matrix factorization of time-varying feature matrix. a** The feature matrices supplied to the NMF ($V$ matrix), and the resultant factorized matrices ($H$ matrix showing temporal expression of six factors; $W$ matrix showing factor weights for six factors). The ictal onset transition can be seen in most of the feature matrices, and occurs around the time factor 2 is expressed most highly. **b** Example factor weights are illustrated for LFP raw power, and calcium fluorescence in the sequence in which they are most highly expressed. Both show sequential increases in combined overall weights, with a maximum in the bottom left quadrant of the matrix, corresponding to the contact nearest the drug application site. **c** Factorized edge weights can be used to reconstruct frequency band-specific subnetworks of which the high gamma-derived subnetworks are shown here. These show a decoupling at seizure onset, with subsequent increases in coupling.

these univariate measures, NMF also yields µECoG-derived pairwise dynamic functional connectivity factors, or "subgraphs" (Fig. 4c). These are modules of functional connectivity whose variable expression over time approximates the full scale original dynamic connectivity matrix in linear supposition[38]. Their temporal progression also tracks transition into the seizure, and further evolution. For example, high gamma connectivity is lowest for factors expressed during seizure onset; and higher for factors expressed in the established seizure, when the ictal core occupies the majority of the electrode grid (Fig. 4c).

The time-varying expression of the six factors further identifies a sequence of state transitions: periods that are characterized by expression of just a single factor alternate with periods characterized by mixed expression of multiple factors, indicating a gradual transition between states (Fig. 4a). Thus, the factor loadings allow us to track transitions through a reduced dimensionality feature space,

and to identify key transitions between transiently expressed states. It is of interest to determine whether these state transitions could be identified without the calcium imaging data, which captures a clearer spatial evolution of activity at slower time scales than the electrophysiology. To address this question, we perform the same analysis separately with calcium imaging features, or with electrophysiological data features alone. We observe that the state transitions are not detectable in the electrophysiology-only analysis. The timing of state changes appears to be preserved in the calcium-only analysis, although the dynamics of individual factor expression differs from the combined analysis (Supplementary Fig. S6). These findings indicate that identifying the sequential latent factors reported here requires the inclusion of calcium imaging data.

Using NMF, we derive a low-dimensional state space representing ictal onset. We depict this state space as a 3-D plot for the three factors that are predominantly expressed early in the seizure (Fig. 5a, b). Next, we depict the 3-D state space of three factors that are predominantly expressed later during the established seizure (Fig. 5c, d). Throughout the recording, transitions between factors are accompanied by smooth slow increases in the spatial extent of the recruited seizure core, as defined by imaging features. In Fig. 5b, d, we show data segments that most closely correspond to each factor, by selecting time windows around the maximum expression of a given factor. These segments show distinctive patterns of specific dynamic features: a gradual increase in the spatial distribution of the calcium signal, changes in ictal discharge morphology and frequency, and strong initial synchronization with subsequent desynchronization in the between-channel functional network.

**State transitions are associated with changes in ictal discharge electrophysiology.** Next, we explore whether features which are in principle measurable in clinical ECoG recordings change quantitatively between seizure states identified from the multimodal NMF analysis. The most prominent features of abnormal cortical dynamics observed in clinical seizure recordings are often ictal epileptiform discharges, which are present in our μECoG recordings (Fig. 6a). For this analysis, we were interested in quantifiable features of ictal discharges that (i) do not depend on the spatial resolution afforded by μECoG (e.g., excluding between-electrode correlation measures), and (ii) were not already included in the NMF analysis (e.g., excluding discharge amplitude). We identify two features that fulfill these criteria, which we hypothesize could act as indicators for ictal state transitions: ictal discharge frequency and ictal discharge stereotypy. We evaluate both features in 3 s nonoverlapping sliding time windows that are labeled according to the factors maximally expressed during that time window. We then compare median values of discharge frequency and between-discharge correlation, as a measure of discharge stereotypy, for temporally adjacent states (Fig. 6b, c).

This analysis reveals contrasting changes in discharge frequency and discharge stereotypy. Discharge frequency monotonically increases from ictal onset, with increasing frequency at each state transition, and reaches a maximum at the end of the recording. However, discharges are most stereotyped early in the seizure, and become most variable in the established seizure around the time windows characterized by high expression of factor 5, before returning to a more stereotyped state toward the end of the recording.

## Discussion

To truly harness the combined advantages of electrophysiology and optical imaging, it is necessary to record and image in the same locations simultaneously. While insights have been gained by utilizing single micropipette local field potential (LFP) recordings in locations proximal to the imaging region[4], these studies do not offer spatial information about the propagation of high-frequency ictal activity, which requires the use of multielectrode arrays. In recent years, marked progress has been made in realizing transparent microelectrode array technology, initially using graphene and more recently utilizing other materials[26]. One of the key challenges in developing transparent microelectrodes has been a trade-off between optical transparency and electrode impedance. For example, the impedance of graphene electrodes, which are subject to graphene's fundamental quantum capacitance limit[39], can be reduced by electrodeposition of platinum nanoparticles on the surface[39], although at the expense of optical transparency. In this study, we have optimized the process for fabricating transparent graphene microelectrodes by utilizing a bubble-transfer method to deposit clean and defect-free graphene layers[40], a layer-by-layer $HNO_3$ doping procedure[41], and sequentially stacked layers of graphene[42]. This optimization resulted in electrodes with a combination of high optical transparency and low impedance, which is the best reported so far for graphene-based transparent electrodes of this size.

Electrophysiological signals detected with μECoG are extracellular in origin, and this allows mixing of signals from different spatial origins and cell types through volume conduction. Thus, even with the most advanced high-density surface μECoG arrays, the effective spatial resolution is limited not only by the electrode spacing, but also by the contributions of volume conduction and the unknown cellular origins of the recorded signal. In contrast, calcium imaging relies on intracellular changes in calcium concentrations which can be more precisely localized to their cellular origins. Furthermore, calcium indicators can be genetically targeted to provide access to precisely isolated signals from specific neuronal subpopulations in a way that is not currently achievable by electrophysiology[43,44].

We used widefield calcium epifluorescence imaging to offer a broad field of view encompassing the entire area covered by the 16-channel graphene array. Widefield imaging combined with somatically expressed calcium indicators is more sensitive to dendritic synaptic currents than individual action potentials, giving a readout of population activity rather than individual neuronal firing. For epileptic seizures, which emerge at the population level, this resolution is an appropriate spatial scale for subsequent analysis. However, some phenomena that characterize seizure activity (e.g., HFOs) cannot be captured with calcium-based methods at all, given the slow dynamics of both calcium transients and the fluorescent calcium indicators. Therefore, our approach allowed us to analyze the complex spatial patterns of ictal wavefront progression over a millimeter-scale cortical area in combination with spatially distributed measures of electrophysiology. In contrast to previous multimodal studies which have utilized cellular resolution multiphoton imaging, our approach also offers a much larger field of view. To observe the contributions of excitatory neurons to ictal dynamics, in this study, we utilized an animal model expressing GCaMP6s in excitatory pyramidal neurons. In the future, our approach can be extended to study the contributions of inhibitory interneurons. This direction would be particularly valuable for shedding light on the delicate balance between excitation and inhibition, and its role in seizure propagation, which is currently a subject of much debate[5,7].

Seizures are characterized by abnormalities in neuronal dynamics spanning across spatial and temporal scales. There is an extensive body of literature mapping these phenomena using a variety of recording setups in a range of different model systems[5,7]. Convergent evidence from these studies suggests that at the onset of a focal seizure, the cortical sheet organizes into

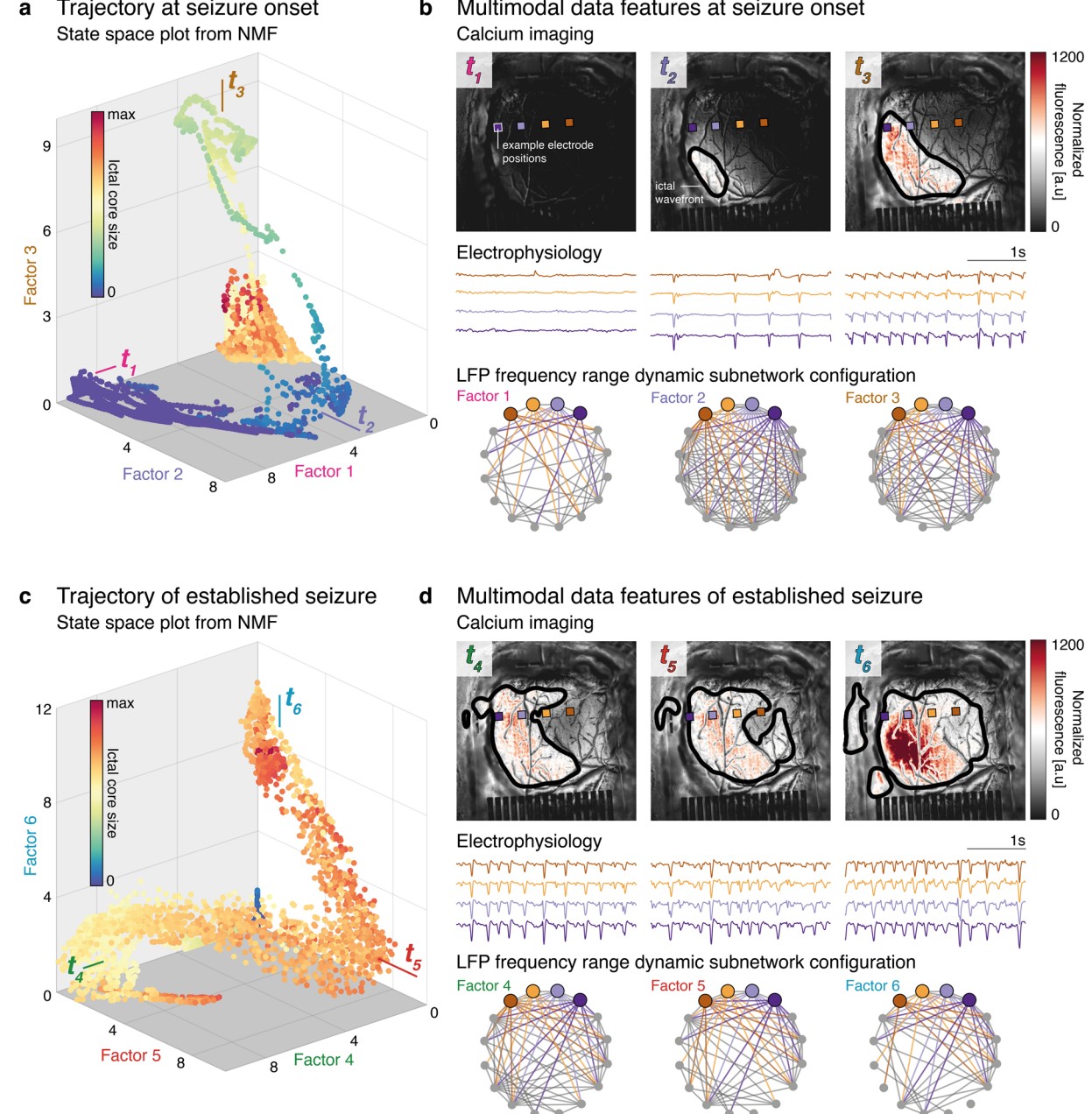

**Fig. 5 Spatiotemporal trajectories during induced epileptic seizure. a** Weighting along three factors showing high loadings early during the seizure, with each point colored by the cortical surface area recruited to the size of the ictal core estimated from the number of suprathreshold pixels on calcium imaging. Seizure onset is associated with a smooth trajectory through this state space. **b** Three time points are sampled based on the extrema of the state-space trajectory, showing normalized calcium fluorescence, raw electrophysiology, and dynamic subnetwork configuration. **c** Analogous to the seizure onset, this plot illustrates the transition through state space based on three factors with maximum activation late in the seizure. **d** Data for three time points late in the seizure are shown, and are associated with gradual μECoG desynchronization along with continuing spatial increase of the seizure core on calcium imaging.

distinct territories characterized by particular dynamics signatures[9]. The recruited, intensely active ictal core is separated by an ictal wavefront from the surrounding cortex (the ictal penumbra) that may be propagating ictal discharges, but lacks intense single neuron firing[1,7]. These distinct seizure territories are best characterized by microelectrode arrays that can record invasively from electrodes piercing pial layers and recording within cortical lamina, but evidence has also arisen from imaging acute seizure-like events in vitro[7].

In our analysis, we quantitatively combine surface μECoG recording of ictal onset with more spatially resolved calcium

mesoscopic imaging, and find that we can identify signatures of these distinct seizure territories in our acute in vivo model of epileptic seizures. We further replicate previous observations of ictal dynamics unfolding on multiple temporal scales: we observe fast spread of individual ictal discharges in both calcium imaging and electrophysiology, as well as the slow, wave-like recruitment of the cortical surface into an ictal core. Similar findings have previously been described based on a variety of electrophysiological features (including multiunit activity[45]), as well as demonstrated in neural field models of cortical physiology[46].

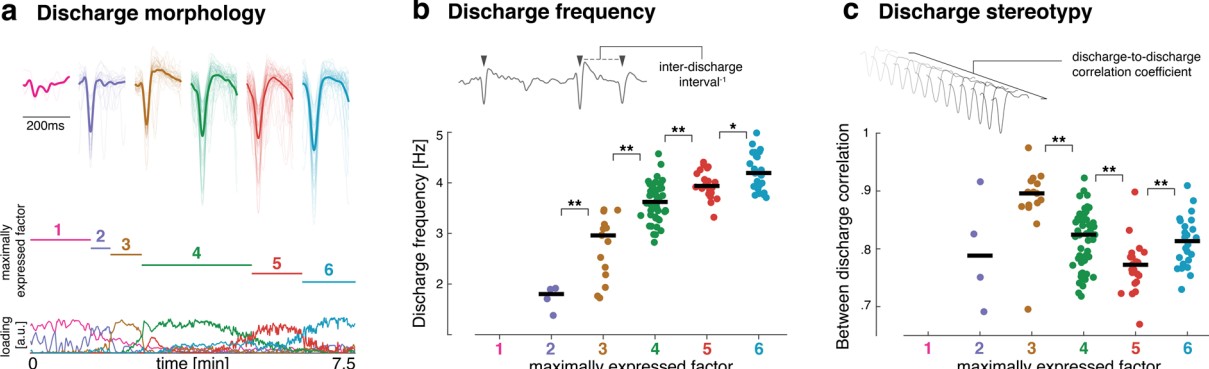

**Fig. 6 Electrophysiological changes associated with ictal state transitions. a** Peak-centered averages of ictal discharges are shown for all discharges detected in a 10 s time window around the peak expression of each of the six sequential factors derived from NMF. Note that for visualization purposes, we show averages at a lower detection threshold, meaning that some fluctuations in the baseline are shown for comparison. For subsequent panels, thresholds were selected to not include baseline fluctuations in the detected discharges. **b** Ictal discharge frequency was estimated for nonoverlapping 3 s sliding windows labeled according to the factor that was maximally expressed during the time window. This shows progressive increases in ictal discharge frequency for the duration of the seizure. **c** Stereotypy of ictal discharges was measured by calculating for each channel the between-discharge correlation for all discharges identified within a 3 s sliding window. This shows that discharges are very stereotyped early in the seizure and become most heterogeneous around the expression of factor 5. Across panels $n = 121$ ictal discharges (mean 24 per factor, range 7–52).

While previous studies have demonstrated simultaneous electrophysiology and imaging, none has so far addressed the challenges that arise in combining and interpreting quantitative features from multiple modalities that operate on different spatial and temporal scales. Dimensionality reduction techniques, such as NMF, are frequently applied to neurophysiological data to extract low-dimensional information from high-dimensional signals, and are thus relevant for the multimodal data on which we report in this study[37]. NMF provides linear representations of nonnegative data, and the algorithm applied here contains a sparsity constraint, allowing us to identify low-dimensional factorizations of the data that contain only a limited number of active contributors to each factor[47]. To apply NMF to multimodal data, modality-specific data matrices are concatenated into a single matrix and aligned in time. This procedure allows us to utilize the high temporal resolution of electrophysiological data by calculating key quantitative features that have been associated with distinct ictal territories—such as phase-locked high gamma. Such features, while requiring high temporal resolution to detect, evolve slowly enough in time to be related to the slower dynamics identified from calcium imaging. The NMF analysis then provides us with temporal loadings on sparse factors that span multiple imaging and electrophysiological features, fully leveraging the strengths of both recording modalities.

Using this approach, we demonstrate that the ictal onset transition—defined by changes in the ongoing oscillatory dynamics of electrophysiological recordings—can be factorized using a data-driven approach into a sequence of states that evolve on the time scale of minutes. Identification of these spatio-temporal patterns is made possible by the combination of recent advances in both transparent electrode fabrication and analytical techniques for processing neuronal data. The framework demonstrated here, enabled by our transparent graphene array technology, allows characterization of ictal phenomena observed with multimodal experimental setups, and helps quantitatively integrate information from concurrent calcium imaging and µECoG recordings.

Recording seizures with multimodal combined µECoG and calcium imaging will not be achievable in human patients in the near future. However, there is already evidence that experimental high-resolution recordings can aid in the identification of novel biomarkers with translational potential. For example,

microelectrode mapping of seizure territories has led to the identification of phase-locked high gamma as a potential biomarker for the ictal core and the seizure onset zone, even on intracranial macro-electrodes[5,48].

In our analysis, we similarly illustrate how high-resolution mapping only achievable in experimental settings might be used to provide new hypotheses that could be testable in clinically achievable recordings. We show that calcium imaging provides data features that separate distinct states within a single seizure, even where aggregate electrophysiological features suggest only a monotonic, gradual change. We harness information on the putative timing of state changes identified as changes in expression of NMF factors. We achieve this by exploring complementary features in the electrophysiological recordings, through which we can identify putative electrophysiological correlates of ictal state changes. One example of such a putative correlate is a gradual loss of local synchrony with progressive enlargement of the seizure core, which can be seen in the changes to the dynamic subnetwork graphs moving from factor 2 to factor 6 (Fig. 5b, d). Similarly, the increasing variability of ictal discharges during the established seizure (Fig. 6c) may be a marker of a specific ictal state, which is potentially measurable with macroscopic clinical ECoG electrodes.

Such changes may be quantitatively subtle, but informative with regard to the underlying dynamical system generating ictal dynamics. If seizures are considered a pathological stable oscillation around a limit cycle[49–51], reductions in stereotypy of discharges may indicate that the system is approaching points of instability where closed-loop interventions may be most successful. These neurophysiological changes, while important, may not themselves be quantitatively sufficient to allow separation into states from dimensionality reduction approaches, as presented here in neurophysiological data alone. However, as demonstrated through our analysis, they may be identifiable from combined analysis with modalities that are more sensitive to dynamics of distinct seizure territories, such as calcium imaging.

Characterizing individual seizures in such detail is important for several reasons: markers associated with distinct spatial territories may guide resective epilepsy surgery and identifying dynamically distinct states within a seizure may enable effective targeting of stimulation for seizure abatement, harnessing closed-loop stimulation approaches for the treatment of seizures. Thus,

deriving candidate biomarkers from experimental animal studies is an essential step in identifying novel treatment strategies for currently untreatable epilepsies.

While the graphene electrode technology presented here enables recording of fast ictal dynamics, we were not able to resolve multiunit activity (MUA) in this study due both to the size of the electrodes (50 μm × 50 μm) and their placement on the cortical surface (relatively distant from the cell bodies which lie in deeper cortical layers). Previous studies have reported successful recording of MUA from the cortical surface[52], when the electrodes are scaled down to cellular dimensions (i.e., ≤30 μm in diameter); however, the high impedance of current transparent microelectrodes has largely prevented the realization of functional electrodes at these cellular spatial dimensions. Further progress in transparent electrode technology may enable high-fidelity recording from electrodes at this scale in the future. Such progress may be achievable by exploring other materials systems, which have shown promise for optoelectronics and may offer higher conductivity than graphene, such as MXenes[53,54]. Access to MUA activity can also be achieved by considering alternative implantation strategies that place the graphene electrodes closer to cell bodies. Such strategies could include brain slice studies or approaches where the array is implanted over deep brain structures, such as the hippocampus, utilizing an implanted cranial window[55].

The seizure model used in this study is a bath application of 4-AP, which offers experimental control and the ability to reliably induce acute seizures for validating our methodology and analytical methods. Yet, the model also has some limitations that are worth noting. In particular, it is challenging to discern the effects of drug diffusion on the spatial propagation of ictal activity, and the pharmacologically induced seizure model may not replicate ictal dynamics that occur in spontaneous seizures. While offering less experimental control, a chronic seizure model in which mice experience spontaneous seizures following drug-induced status epilepticus may offer a more naturalistic view of ictal dynamics[56].

In terms of analysis, the methodological choices make certain assumptions. Applying dimensionality reduction techniques (such as NMF) to high-dimensional neuronal data assumes that key information is contained in large signal changes affecting a high proportion of recording channels. The representational accuracy of the low-dimensional approximation of the full dataset can be arbitrarily increased by increasing the number of factors or components, and a trade-off between accuracy and interpretability must be made. Furthermore, NMF assumes that the dynamics emerge from variable expression of spatially invariant factors, rather than changes in the spatial factors themselves. Other conceptual approaches or interpretations may prove more useful for specific cases.

Multimodal recording and analysis methods such as the ones introduced here improve our ability to quantify and ultimately understand the dynamics of epileptic activity in local microcircuits, and brain-wide epileptogenic networks. Concurrently recorded multimodal data may be used to address biological questions that are not tractable with a single method approach. For example, genetically targeted calcium imaging of inhibitory neuron activity during concurrent measurement of the μECoG may enable separating the contributions of excitatory and inhibitory cell types to the ECoG signal. Combining μECoG with imaging of cortical vasculature using injected fluorescent probes may also enable linking neural activity patterns to changes in blood flow, offering a direct proxy for functional magnetic resonance imaging studies[57].

Increasingly, through resective epilepsy surgery and laser ablation for example, patients are being offered interventions that aim to disrupt epileptogenic networks, in order to reduce or stop their seizures. These therapies are currently driven by a crude understanding of cortical dynamics at the scale of ECoG signals. This understanding is based on coarse physical models, such as mesoscale neural masses representing large homogeneous populations of different neuronal subtypes[58], and equivalent current dipoles representing the electromagnetic spread of single source signals[59]. The sequence of spatiotemporal patterns constituting distinct states in the ictal onset reported here all occur at a spatial scale that would classically be regarded as a single source in most ECoG models. We now have the technological ability to acquire data at unprecedented temporal and spatial resolutions, and have the computing power and analysis methods at hand to identify from these complex data the key features that capture the dynamics at play. Through high-fidelity recordings in experimental systems, we will be able to inform the development of hypotheses and models that may more accurately explain the phenomena observed in clinical recordings in human patients.

There is an urgent unmet need for novel therapies for patients who cannot currently benefit from resective surgery or anti-epileptic medication. Treatment approaches such as responsive neuromodulation using electrically stimulating devices[60,61], or novel optogenetic closed-loop control strategies[62] are currently already in clinical use or in development. Yet such modulation-based treatments require a detailed characterization of state transitions in local epileptogenic circuits, as well as models of how interventions may disrupt these transitions. The integrated recording and analysis approach presented here is one strategy to quantify these state transitions utilizing optical and electrophysiological measurements. As such, we provide essential tools for epilepsy research to explore novel treatments in experimental systems.

## Methods

**Graphene device fabrication**. Three-inch silicon wafers were coated with 4 μm parylene-C using a chemical vapor deposition process (SCS Labcoter 2 Parylene Deposition System). Following photolithography, 10 nm titanium (Ti) and 100 nm gold (Au) were evaporated onto the wafer using an electron-beam deposition process, and lift-off techniques were used to pattern the metal to form traces and connection pads. Monolayer graphene was transferred from a copper foil to the wafer using a wet bubble-transfer method consisting of the following steps: spin-coat 950 PMMA A4 on graphene/copper foil; bake at 100 °C for 2 min; apply 20 V potential between tweezers and stainless steel counter electrode in 1 mM NaOH solution, and grip copper foil in tweezers while slowly submerging into NaOH, releasing graphene/PMMA from foil; transfer graphene/PMMA to three water baths for cleaning; transfer graphene/PMMA onto wafer substrate and let air dry for 2 h; bake wafer at 150 °C for 2 min, then remove PMMA with acetone, IPA, and deionized (DI) water rinse. Following each graphene transfer to the wafer, the graphene was doped by soaking the wafer in 75% $HNO_3$ in DI water at room temperature for 15 min and subsequently dried with an $N_2$ gun. $SiO_2$ (30 nm) was then deposited onto the wafer using e-beam deposition to protect the graphene from damage during subsequent reactive ion etching (RIE) steps. The $SiO_2$ and graphene were patterned using photolithography and RIE to form the electrode array and traces, such that graphene traces overlapped Ti/Au traces. A second 4 μm layer of parylene-C was deposited through chemical vapor deposition, and subsequently patterned through photolithography and RIE to form the outline of the device, and to open the graphene electrode sites and Ti/Au contact pads. The protective $SiO_2$ layer was removed from the graphene contacts by immersing the wafer in 1:6 buffered oxide etchant for 30 s. Devices were released from the wafer and Kapton tape was applied to the back of the Ti/Au bond pad region to ensure proper thickness and stiffness for insertion into a zero insertion force (ZIF) connector.

**Electrochemical characterization**. Electrochemical impedance spectroscopy (EIS) measurements were made using a Gamry Reference 600 potentiostat/galvanostat/ZRA in standard three-electrode configuration in 10 mM phosphate-buffered saline solution (pH 7.4). A graphitic carbon rod was used as the counter electrode, and an aqueous (3 M KCl) Ag/AgCl electrode was used as the reference. EIS measurements were taken between 1 Hz and 1 MHz using a 20 mV rms AC voltage. Cyclic voltammetry measurements were also taken using the Gamry Reference 600 from −0.8 to 0.8 V, with scan rates of 200, 300, 400, and 500 mV/s and a potential step of 5 mV. The cathodal charge storage capacity ($CSC_C$) was calculated by integrating the area under the average CV curve and dividing by the sweep rate (50 mV/s).

**Ultraviolet–visible spectroscopy**. Transmission mode UV–vis spectroscopy was obtained using a Perkin-Elmer Lambda-950 spectrophotometer over wavelengths from 300 to 1000 nm. Spectra were taken for the interelectrode space (consisting of top and bottom parylene-C layers only), for a graphene electrode contact (consisting of the bottom layer parylene-C and the graphene layer), and for a graphene trace (consisting of top and bottom parylene-C layers and graphene layer).

**In vivo imaging and electrophysiological recording**. All experiments were performed under a protocol approved by the Children's Hospital of Philadelphia Institutional Animal Care and Use Committee. Briefly, 6–8-week-old Thy1-GCaMP6 (Jax #025776, Jackson Laboratory) mice were anesthetized with isoflurane (0.5–1.5%) in oxygen. Using a micro-drill, a hole was made for inserting a stainless steel skull screw with a silver ground wire. A metal headplate with $6 \times 6$ mm² square imaging well (Narishige USA) was fixed to the skull overlying the somatosensory cortex with dental acrylic (Ortho-Jet). A rectangular craniotomy was drilled (right hemisphere, center at 2 mm posterior and 2 mm lateral of bregma). The exposed cortical surface was kept wet by ACSF containing the following (in mM): 155 NaCl, 3 KCl, 1 MgCl₂, 3 CaCl₂, and 10 HEPES. After the craniotomy surgery, aesthetic was switched to ketamine/xylazine (100/10 mg/kg) and the headplate was attached to a small imaging platform (Narishige USA) and mounted on an $x$, $y$-translational stage of an upright microscope (Olympus BX-61). The graphene electrode array, a custom-connector, and the headstage (SmartLink16, NeuroNexus) were attached to a stainless steel bar, which was then attached to a micromanipulator. The graphene electrode array was gently placed on the cortical surface. Signals from the graphene electrode array were sampled at 5 kHz via the headstage and the controller (SmartBox, Neuronexus), and the data were saved to a PC. The microscope was equipped with blue LED (470 nm, Thorlabs), a filter cube (49002, Chroma), and sCMOS camera (Orca Flash 4 V2, Hamamatsu). The imaging data was captured in TIF format on another PC, and timing of electrophysiological recording and imaging was synched by a TTL signal. For epifluorescence imaging, excitation light was continuously illuminated and fluorescence images were captured at a constant acquisition rate of 10 Hz. Baseline activity was recorded for 20 min prior to seizure induction. Local seizure induction was achieved via bath application of 100 µL of 4-AP in ACSF (30 mM) on the exposed cortical surface adjacent to the graphene electrode array.

**SNR calculation**. To compute SNR on each electrode, the mean amplitude of eight ictal spikes that occurred during a 4-second window during seizure onset was divided by the RMS of the signal during a 3-second window of pre-seizure baseline activity.

**Dynamic feature extraction**. Quantitative analysis was performed on custom scripts on MATLAB R2019a; all scripts to reproduce analysis figures are available online on https://github.com/BassettLab/Graphene-Electrode-Seizures.

For this analysis, we first extracted quantitative features from µECoG recordings that dynamically evolve at a time scale similar to that evident in the features of calcium imaging. Specifically, we estimated time-varying phase-locked high gamma amplitude, frequency band-specific power in two frequency bands, and sliding window dynamic functional connectivity.

*Phase-locked high gamma*. We estimated phase-locked high gamma amplitude $\hat{a}_{high}^{\Phi}$ following prior work[5]. Briefly, we applied the Hilbert transform separately to the data recorded at each microelectrode contact, after filtering each into two distinct frequency bands: a low-frequency band (1–40 Hz) and a high-frequency gamma band (80–250 Hz). From the Hilbert transform, we were then able to calculate low-frequency instantaneous phase $\Phi_{low}$, and high-frequency instantaneous amplitude normalized by the average band-passed amplitudes at baseline $a_{high}$. From a second Hilbert transform, we derived the instantaneous phase of the slowly time-varying instantaneous amplitude of the high-frequency content $\Phi_{a_{high}}$. From these values, we calculated the phase-locked high gamma amplitude as a measure of normalized high-frequency amplitude weighted by the mean cross-frequency band phase correlation. Specifically, we calculated this variable within the sliding window ranging from sample $n = 1$ to sample $n = N$:

$$\hat{a}_{high}^{\Phi} = \left| \frac{1}{N} \sum_{n=1}^{N} a_{high} \exp(i(\Phi_{low}^n - \Phi_{a_{high}}^n)) \right| \tag{1}$$

We calculated this variable through sliding windows of length 3 s and then then interpolated linearly to a full sampling frequency of 10 Hz to enable cross modality analysis with the imaging features.

*Frequency band-specific power*. We calculated frequency band-specific power using the Hilbert transform on time series filtered into low- (1–40 Hz), and high- (80–250 Hz) frequency components, tracking evolution of most of the classically visible frequency ranges, and of HFOs separately. We estimated band-specific power in 3 s nonoverlapping sliding windows, before interpolating to the 10 Hz sampling frequency of the calcium imaging in order to track slow modulations of frequency power.

*Functional connectivity*. To estimate functional networks, we calculated pairwise correlations between individual microelectrode contacts. Full correlation matrices were estimated for sliding 3 s windows, edges with correlation values of <0 were reset to 0, and edge-wise temporal evolution was interpolated to the sampling frequency of the calcium imaging at 10 Hz.

We also derived normalized measures of fluorescence intensity during the seizure, as well as a measure of each pixel's proximity to a putative ictal wavefront. Imaging yielded raw epifluorescence image sequences for the baseline period, and from the period following the application of the chemoconvulsant 4-AP. We used the baseline period to quantify pixel-wise mean fluorescence, and fluorescence intensity variability (estimated as the standard error of the mean). From the baseline mean and standard error maps, we then calculated a $t$ statistic map for each image in the sequence following chemoconvulsant exposure, where

$$t = (x - \mu_{BL})/\sigma_{\bar{x}_{BL}} \tag{2}$$

with $x$ being the sample value, $\mu_{BL}$ the baseline mean, and $\sigma_{\bar{x}_{BL}}$ the baseline standard error of the mean.

In order to coarse-grain the data, we applied a two-dimensional gaussian filter with a width of 15 pixels to each $t$ map in the sequence, and binarized the resultant images at a manually selected threshold that would first be crossed at the time of visually apparent electrographic seizure onset. We detected the edge of these binarized maps at each time point and convolved the edges with a two-dimensional Gaussian kernel with a width of 15 pixels, which yielded a measure of proximity of each imaging pixel to the putative ictal wavefront.

Finally, we sampled both raw fluorescence, and proximity to the putative wavefront from the imaging pixels underlying each of the 16 microarray contacts, generating 16 time series for both measure, which were then used in the subsequent analysis.

**Nonnegative matrix factorization**. We used NMF on a multimodal feature matrix containing the time-varying features estimated above in order to identify a subset of composite factors, whose time-varying expression could recapitulate the multimodal feature evolution observed in the data. NMF will decompose any $n \times t$ feature matrix $V$ (where $n$ is the number of features, and $t$ is the number of temporal samples) into a set of $k$ factors (i.e., sets of features encoded in the $W$ matrix) and their temporal evolution (encoded in the $H$ matrix), so that the full matrix $V$ can be approximately reconstructed from these factors (i.e., $V \simeq WH$). This reconstruction is usually found by solving the optimization problem

$$\min_{W,H} f(W, H) = \frac{1}{2} ||A - WH||_F^2, \tag{3}$$

where all elements of $A$, $W$, and $H$ are positive values.

In terms of network analysis, these approaches have been previously used to decompose dynamic functional connectivity into sets of variably expressed subnetworks and track their temporal evolution in human neuroimaging[27–29]. Here, we extend the application of the approach to integrate data features across modalities and feature types. In contrast to the original formulation of the method, the algorithm applied here uses additional sparsity constraints for the $H$ factor[35,36]:

$$\min_{W,H} \left\{ ||A - WH||_F^2 + \eta ||W||_F^2 + \beta \sum_{j=1}^{n} ||H(:, j)||_1^2 \right\}, \tag{4}$$

where $\eta$ is a small positive parameter to relatively suppress the contribution of $W_F^2$ and $\beta$ is a positive regularization parameter to balance the accuracy/sparseness approximation.

The full feature matrix $V$ was concatenated from channel-specific time series of broadband LFP (1–40 Hz), and high gamma (80–250 Hz) band power, normalized fluorescence amplitude, and wavefront proximity estimates, as well as the $n_c(n_c - 1)/2$ unique edge weights unfolded from the symmetrical correlation-based $n_c^2$ functional connectivity matrix, where $n_c$ is the number of channels.

**Ictal discharge analysis**. We quantified between-state differences in features of ictal discharge. For this, we divided the whole seizure recording included in the NMF analysis into 3 s long nonoverlapping time windows, labeled according to the factor that was maximally expressed during each window. We $z$-transformed data for each time window using baseline data to calculate normalizing mean and standard deviations individually for each channel. In the resultant $z$-normalized time series, we identified putative ictal discharges as negative deviations of an amplitude of $z > 3.9$ and a width of at least ~30 ms. These parameters were identified by a clinician with training in epilepsy (RER) visually confirming that the detected events were ictal in nature, and no baseline events were detected (for visualization purposes in Fig. 6a, the thresholds were slightly relaxed in order to allow inclusion of some baseline detections—but those were not included in subsequent statistical analysis).

We calculated the average inter-discharge interval for each time window as an inverse measure of spike frequency. We also estimated the correlation of all ictal discharges within a time window. For this, we segmented out ictal discharges from 50 ms prior to the peak amplitude to 150 ms after peak amplitude, and calculated Pearson's correlation coefficient between each pair of these segments within time windows and channels. We then averaged all correlation coefficients to estimate a

single mean correlation value per 3 s window, representing ictal discharge stereotypy. We compared each pair of sequential states (defined by their factor expression) by Wilcoxon rank-sum comparison of discharge frequency and stereotypy for each group of time window associated with either state.

**Citation diversity statement**. Recent work in neuroscience and other fields has identified a bias in citation practices such that papers from women and other minorities are under-cited relative to the number of such papers in the field[63–68]. Here, we sought to proactively consider choosing references that reflect the diversity of the field in thought, form of contribution, gender, and other factors. We used automatic classification of gender based on the first names of the first and last authors[63,69], with possible combinations, including male/male, male/female, female/male, and female/female. Excluding self-citations to the senior authors of our current paper, the references contain 50% male/male, 28% female/male, 13% female/male, and 9% female/female categorization. We look forward to future work that could help us to better understand how to support equitable practices in science.

**Statistics and reproducibility**. To illustrate the analyses enabled by the recording setup demonstrated here, we analyzed a single example seizure in detail. To evaluate relationship between two continuous variables (e.g., normalized μECoG broadband power, and normalized fluorescence), we calculated the Pearson's correlation coefficient and the associated $p$ value calculated by transforming the correlation to create a $t$ statistic with $N-2$ degrees of freedom, where $N$ is the number of samples tested (implemented in the MATLAB `corrcoef` function). Relationships between medians of grouped variables were compared using the Wilcoxon rank-sum test as implemented in the MATLAB `ranksum` function. Where applicable, $p$ values of <0.05 were considered statistically significant.

**Reporting summary**. Further information on research design is available in the Nature Research Reporting Summary linked to this article.

## Data availability

The electrophysiology recorded on the graphene electrode μECoG, as well as the calcium fluorescence imaging data are publicly available on https://doi.org/10.6084/m9.figshare.13007840 (ref. [70]). Figures containing raw data from these datasets include: Figs. 2, 3, and 5.

## Code availability

All code used to generate the results in this work are publicly available on: https://doi.org/10.5281/zenodo.4050520 (ref. [71]).

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

## Acknowledgements

This material is based upon work supported by the National Science Foundation Graduate Research Fellowship Program (DGE-1845298 to Driscoll, Murphy) and was carried out in part at the Singh Center for Nanotechnology, which is supported by the National Science Foundation through the National Nanotechnology Coordinated Infrastructure Program (NNCI-1542153). Any opinions, findings, conclusions, or recommendations expressed in this material are those of the authors and do not necessarily reflect the views of the National Science Foundation. This work was also supported by the National Institutes of Health (R21-NS106434 to F.V. and H.T.), the National Institute of Neurological Disorders and Stroke (R01 NS099348 to B.L. and D.S.B.), the Wellcome Trust Sir Henry Wellcome Fellowship (209164/Z/17/Z to R.R.), and the Citizens United for Research in Epilepsy Taking Flight Award (F.V.). A.T.C.J. acknowledges support from NSF through MRSEC DMR-1720530.

## Author contributions

N.D., F.V., and H.T. designed the research. N.D. designed and fabricated the graphene arrays and custom connectors. H.T. performed the animal surgeries and together with N.D. conducted the imaging and recording experiments. N.D. and R.R. performed the in vivo data analysis and wrote the manuscript. B.M. optimized the nitric acid doping of graphene, performed additional device fabrication, and conducted in vitro characterization of the graphene devices. R.V. and O.D. provided high-quality monolayer graphene with guidance from A.T.C.J. D.S.B. contributed computational consultation and edited the paper. K.A.D. and B.L. provided seizure annotations and clinical epilepsy guidance. A.A. contributed computational consultation.

## Competing interests

The authors declare no competing interests.
