## [Peer Review File · Communications Biology]

Reviewers' comments:

Reviewer #1 (Remarks to the Author):

The manuscript by Driscoll et al. designed and fabricated a transparent electrode array using HNO₃ doped graphene, and achieved simultaneous electrophysiological recording and calcium imaging with this electrode array. The novelty and significance of this work mainly comes in two aspects: first, although transparent graphene electrode array had been reported for simultaneous electrophysiological recording and calcium imaging, the current work expanded its application to seizure dynamics investigation. Second, the authors developed a non-negative matrix factorization method to combine high density optical and electrophysiological datasets. This method could be extended to other studies. This work described here shows promise in neurology studies by creating a multi-modal neural interfacing including optical imaging and electrophysiological recording. However, there are still some questions that need to be addressed before publication:

1. From the data shown in Figure 1e, the impedance varied from $\sim 0.25\text{kohm}$ to $\sim 2\text{kohm}$. Will the impedance difference lead to a different signal-to-noise ratio (SNR), and thus affect the signal, which would give rise to some false propagation results? For example, is there a difference in SNR between the blue electrode site and the red electrode site?

2. Line 93 "microscale neurophysiological markers of the so-called 'ictal wavefront' spread locally at much slower speeds, typically not exceeding $\sim 5\text{ mm/s}$."

Figure 3e, "the speed of fluorescence map: 12.5 s per frame."

Although the propagation speed is less than 5mm/s , it also spread over a wide area in the time of 12.5s which was required to take a frame in calcium image. If the propagation process repeats forward and backward within 12.5s, it seems impossible for the calcium imaging to see the changes of the ictal wavefront. This temporal resolution of 12.5 s seems to be inadequate for the seizure dynamics study.

3 Line 160: Why HNO₃ chemical doping can minimize impedance and improve CSC of graphene? And how stable is this improvement? Please provide some insight on the underlying mechanism.

4 Line176-181. Page25 L334-345

Wide-field epifluorescence imaging does have large field of view compared to the two-photon imaging, but with reduced spatial and temporal resolution, as well as reduced penetration depth. Related with question 2 above, please address how the reduced temporal resolution, and penetration depth affect the seizure dynamics study, especially the reduced temporal resolution.

5 Line201-203 Ref36

It has been reported in Ref36 that a large area, high density electrode array can also be used to map electrographic seizures, and investigate the onset and propagation of epilepsy in vivo. Using the high density electrode array, detailed propagation spatial patterns has been recorded in ref 36. How does the multimodal recording using transparent microelectrodes here compared to the high density, large area electrophysiological method in ref36? To the reviewer's understanding, if using high-density electrode array, same results on seizure dynamics can be obtained even without the calcium imaging.

Reviewer #2 (Remarks to the Author):

In the manuscript, Driscoll and colleagues present simultaneous monitoring of epileptic seizures using electrophysiology recordings and fluorescence imaging in anesthetized 4-AP mouse model. This is a highly challenging experimental procedure and to achieve such data, the authors utilized graphene-based surface probes to create a semi-transparent electrophysiologic probe that allows optical imaging through the probe. The main reasoning for such multi-modal measurements is to be able to acquire neural activity at temporal resolution of electrophysiology and spatial resolution of wide field imaging.

Overall, I found that the manuscript contained the necessary experimental data and explanations to support the authors' claims. However, there are a few points that I believe should be addressed to strengthen the manuscript.

- The ECoG array used here is approximately ($1.5 \times 1.5 \text{ mm}^2$, 16 electrodes) while the effective optical ROI is about ($3 \times 3 \text{ mm}^2$) based on Figure 2d. Also, the dorsal surface of a mouse brain is $4 \times 5 \text{ mm}^2$. Considering the capacity of the authors in fabricating high-density and high channel count probes, it would certainly be beneficial to discuss the advantages of the combinatory approach as opposed to increasing the spatial coverage of their minimally invasive electrophysiological probe to have both high spatial and temporal resolution with a single modality.
- Figure 2C: It would be beneficial to have multiple electrophysiological waveform traces to highlight the localized nature of the HFO.
- Figure 2D (right): If I understood this graph correctly, the early time-points (blue) of the seizure have larger spatial extent than later time points (red). Is this a representative example?
- Figure 3E: "we observe a correlation between normalized μECoG broadband power, and local average normalized calcium fluorescence Fig. 3e " would benefit from further explanation and clarification. What is the correlation? Why is the heat map unit normalized frequency and not Hz?
- A major pre-processing step of wide field imaging is dedicated toward extracting vasculature dilation. In this unique experiment, it would be interesting to provide comparison of neural activity patterns to changes in blood flow, which would be a direct proxy for fMRI studies. Although experimentally not necessary, the manuscript would certainly benefit from a discussion surrounding such opportunities in multimodal experiments.

We thank the reviewers for their constructive feedback, which we felt improved the manuscript significantly. In the revisions detailed below, we carefully and thoroughly address the points raised by both reviewers. In the main manuscript file, all changes that have been made are indicated in red type.

Reviewer #1:

The manuscript by Driscoll et al. designed and fabricated a transparent electrode array using HNO₃ doped graphene, and achieved simultaneous electrophysiological recording and calcium imaging with this electrode array. The novelty and significance of this work mainly comes in two aspects: first, although transparent graphene electrode array had been reported for simultaneous electrophysiological recording and calcium imaging, the current work expanded its application to seizure dynamics investigation. Second, the authors developed a non-negative matrix factorization method to combine high density optical and electrophysiological datasets. This method could be extended to other studies. This work described here shows promise in neurology studies by creating a multi-modal neural interfacing including optical imaging and electrophysiological recording. However, there are still some questions that need to be addressed before publication:

1. From the data shown in Figure 1e, the impedance varied from $\sim 0.25\text{k}\Omega$ to $\sim 2\text{k}\Omega$. Will the impedance difference lead to a different signal-to-noise ratio (SNR), and thus affect the signal, which would give rise to some false propagation results? For example, is there a difference in SNR between the blue electrode site and the red electrode site?

The impedance values shown in Figure 1e reflect impedance values averaged across 3 graphene electrode arrays measured in saline solution in a benchtop test. The figure thus offers information from a representative sample of the devices used during the *in vivo* recording experiments. The impedance values shown in Figure 1e range from 551 k Ω to 1830 k Ω . Please note that the scale on the colorbar in Figure 1e is in M Ω , rather than k Ω . Such variability in electrode impedance reflects the challenging nature of fabricating pristine transparent graphene electrodes. However, the variance in our impedance data is in line with what has been shown previously by other groups developing or studying transparent graphene electrodes. For example, D. Ding *et al.* reported impedance values ranging from approximately 750 k Ω to 3000 k Ω across a 16-ch array of 100 x 100 μm^2 transparent graphene electrodes which are 4x larger in geometric area compared to our 50 x 50 μm^2 electrodes.¹

During the *in vivo* recording experiment in contrast to saline solution, we measured significantly higher impedance values on the graphene electrodes. During the seizure onset analyzed in this paper, the *in vivo* impedance values measured on the graphene array ranged from 5.8 M Ω (electrode “c” from Figure 3a) to 10.9 M Ω (electrode “e” from Figure 3a). This result is expected as impedance values measured *in vivo* may be significantly higher than those measured in saline solution due to several factors, including: 1) tissue contact, 2) reduced ionic diffusivity in tissues compared to saline, and 3) the higher impedance of the reference or counter electrode used *in vivo* (here, a skull screw).^{2,3} However, we were still able to obtain signals with high SNR. Previous studies have shown that transparent graphene electrodes record signal with lower noise and higher SNR compared to same-size gold electrodes, even when they have comparable impedance⁴. This phenomenon is attributed to the high capacitance of graphene electrodes, and serves as an example to demonstrate that impedance and electrode area are not the only factors which determine the noise performance of electrodes, as predicted by modelling studies.^{5,6}

To address the question of whether variability in impedance could give rise to false propagation results in our analysis, we computed the SNR for each electrode and plotted these values against the

corresponding *in vivo* impedance. We found no significant relationship between SNR and electrode impedance (linear fit $R^2 = 0.0003$). In fact, we found that the distribution of SNR values most closely mapped to the spatial pattern of seizure onset and propagation, with the highest SNR found on electrode “m” at the bottom left corner, closest to the site of seizure onset. This finding suggests that the amplitude of the recorded seizure activity is closely related to the electrode’s proximity to the seizure focus, and that this proximity-related amplitude scaling dominates the SNR rather than electrode impedance. This phenomenon can also be observed in the selected electrophysiology traces shown in Fig 5b,d, where the amplitude of the ictal discharges scales with proximity to the seizure focus. We have added an additional figure, Supplementary Fig. 4 (**Supplementary Information, p6, line 96-101**) to summarize these findings and demonstrate that variability in electrode impedance did not have an effect on the results of our analysis:

Supplementary Fig. S4 | SNR vs. electrode impedance. **a**, SNR vs. 1 kHz impedance magnitude for each electrode during the *in vivo* recording analyzed in this work. No correlation between SNR and impedance was found, as evidenced by the poor linear fit shown. Note that while the impedance magnitudes measured *in vivo* are significantly higher than those measured in saline, these electrodes still achieve high SNR >5. **b**, Map of *in vivo* electrode impedances across the 16 ch graphene array. **c**, Map of SNR values across the 16 ch graphene array.

We now also include the following paragraph into the main text to summarize the findings of this additional analysis:

Line 193-200: It is also notable that our transparent graphene electrodes recorded seizure activity with high signal-to-noise ratio (SNR) (Supplementary Fig. 4). Despite some variability in electrode impedance across the array during *in vivo* recordings, we found that all electrodes had SNR >5, and that SNR was not in fact correlated with electrode impedance (Supplementary Fig. 4a,b). Instead, the SNR values across the electrode array mapped more closely to the seizure onset pattern, with electrodes closer to the seizure focus showing larger amplitude ictal spikes and thus higher SNR (Supplementary Fig. 4c). While higher impedance electrodes do pick up more 60 Hz noise interference, this artifact is removed in the data filtering step such that electrode impedance does not affect the results of our analysis.

We have updated the methods section to describe the SNR calculations as follows:

Line 572-575: To compute SNR on each electrode, the mean amplitude of 8 ictal spikes that occurred during a 4-second window during seizure onset was divided by the RMS of the signal during a 3-second window of pre-seizure baseline activity.

Additionally, we have updated the mean impedance value reported in the text to reflect an average across a larger number of graphene electrodes measured in saline. The new value reported, 908.2 ± 488 k Ω , reflects an average across 64 graphene electrodes, whereas the previous value reflected an average across 48 electrodes. We additionally report this updated value in an area-normalized format, to enable easier comparison to values across the literature, where the electrode size may vary. These changes were made in the main text as follows:

Line 160-162: The average impedance over 4 such devices (n=64 channels) was 908.2 ± 488 k Ω at the 1 kHz reference frequency (Fig. 1e and Supplementary Fig. S2), which corresponds to an area-normalized impedance of 22.7 ± 12.2 $\Omega \cdot \text{cm}^2$.

2. Line 93 “microscale neurophysiological markers of the so-called ‘ictal wavefront’ spread locally at much slower speeds, typically not exceeding ~ 5 mm/s.”

Figure 3e, “the speed of fluorescence map: 12.5 s per frame.”

Although the propagation speed is less than 5mm/s, it also spread over a wide area in the time of 12.5s which was required to take a frame in calcium image. If the propagation process repeats forward and backward within 12.5s, it seems impossible for the calcium imaging to see the changes of the ictal wavefront. This temporal resolution of 12.5 s seems to be inadequate for the seizure dynamics study.

We apologize if the labelling in Figure 3c was misleading. The image acquisition rate in these experiments was 10 frames per second, as stated in the methods section, and we have now also included this information in the figure legend. In Figure 3c, selected image frames spaced 12.5 seconds apart are displayed in order to show the full progression of the ictal onset and spread. We have updated the figure caption to clarify this point, and have also noted the frame acquisition rate of 10 Hz in the main text (**Line 169-170**). Because the actual frame acquisition rate was 10 Hz, we do in fact have adequate temporal resolution to capture the dynamics of ictal wavefront propagation. The figure caption now reads:

Fig. 3: [...] c, Series of image frames corresponding to the seizure onset shown in panel b. From the full image sampling resolution of 10Hz, frames are sampled here for visualization only every 12.5 seconds (0.08Hz), illustrating the entire progression of seizure onset and propagation.

3. Line 160: Why HNO₃ chemical doping can minimize impedance and improve CSC of graphene? And how stable is this improvement? Please provide some insight on the underlying mechanism.

As described in the Supplementary Information section “Optimization of graphene microelectrode array fabrication”, HNO₃ chemical doping results in the adsorption of electropositive NO₃⁻ groups onto the graphene surface, which results in a p-type or hole doping. Specifically, the HNO₃ molecule physisorbs onto the graphene sheet, without breaking any C-C bonds, and then dissociates into three groups: two radicals NO₂⁰ and NO₃⁰, and a water molecule: $2\text{HNO}_3 = \text{NO}_2^0 + \text{NO}_3^0 + \text{H}_2\text{O}$. The two radicals have a singly occupied state below the Fermi energy of the graphene layer, which allows two electrons to transfer from graphene into these states, creating two holes and causing p-type doping.⁷ This p-type doping results in reduced sheet resistance for the graphene layer, which gives rise to reduced electrode impedance. D. Kuzum *et al.* also showed that HNO₃-doped graphene electrodes have smaller charge transfer resistance and increased capacitance, which contributes to their improved CSC compared to non-doped graphene electrodes.⁴ Regarding the stability of such doping, it is true that the doping effects may wear off over time as the NO₃⁻ groups desorb from the graphene surface. However, in physiological conditions this desorption process is very slow. To our knowledge, no study thus far has characterized

the long-term stability of HNO₃ doping of graphene in physiological conditions. However, we have evaluated the impedance of our electrodes *in vivo* every 30 min for up to 5 hrs, and we did not observe any increase in impedance in that time. Thus, we are confident that the doping was stable over the length of the acute experiments shown in this work. We have added additional details to expand upon the discussion of the underlying mechanism of HNO₃ doping in the Supplementary Information section, as follows:

Supplementary Information, Line 37-43: Exposing graphene to nitric acid (HNO₃) results in the adsorption of electropositive NO₃⁻ groups onto the graphene surface. Specifically, the HNO₃ molecule physisorbs onto the graphene sheet, without breaking any C-C bonds, and then dissociates into three groups: two radicals NO₂⁰ and NO₃⁰, and a water molecule: 2HNO₃ = NO₂⁰ + NO₃⁰ + H₂O. The two radicals have a singly occupied state below the Fermi energy of the graphene layer, which allows two electrons to transfer from graphene into these states, creating two holes and causing p-type doping⁷. This p-type or hole doping has been shown to reduce electrochemical impedance and improve the noise characteristics of graphene electrodes⁷⁻⁹.

4. Line176-181, Page25 L334-345

Wide-field epifluorescence imaging does have large field of view compared to the two-photon imaging, but with reduced spatial and temporal resolution, as well as reduced penetration depth. Related with question 2 above, please address how the reduced temporal resolution, and penetration depth affect the seizure dynamics study, especially the reduced temporal resolution.

The temporal resolution of the widefield fluorescence demonstrated here (10 Hz, as clarified in response to question 2 above) is broadly similar to that achieved by two-photon imaging,^{10,11} depending on the spatial coverage, and it is sufficient to capture the spatial spread of the ictal wavefront, as clarified in response to question 2 above. Epifluorescence imaging does not offer single cell spatial resolution as two-photon imaging does, however for the purposes of the seizure dynamics study herein, imaging the entire cortical area under the electrode array was more important than obtaining cellular resolution within a much more restricted field of view beneath one single electrode. The spatial resolution of the wide-field epifluorescence imaging used in this study was 6.6 μm per pixel, which is sufficient for mapping the spatial complexities of the ictal wavefront progression, as this is a reflection of population activity rather than individual neuronal firing. With regards to penetration depth, while it is true that wide-field epifluorescence has reduced penetration depth compared to two-photon imaging, we are able to capture the activity of the superficial cortical layer, which reflects the firing activity of the layer 2/3 pyramidal neurons captured at the dendrites. In this study, we were interested in mapping the dynamics of seizure spread across the cortical surface, and thus the surface activity visible in epifluorescence imaging is sufficient. We have added additional justification for the use of widefield fluorescence imaging in the main text as follows:

Line 350-358: Widefield imaging combined with somatically expressed calcium indicators is more sensitive to dendritic synaptic currents than individual action potentials, giving a readout of population activity rather than individual neuronal firing. For epileptic seizures, which emerge at the population level, this is an appropriate spatial scale for subsequent analysis. However, some phenomena that characterize seizure activity (e.g. high frequency oscillations) cannot be captured with calcium-based methods at all, given the slow dynamics of both calcium transients and the fluorescent calcium indicators. Therefore, our approach allowed us to analyze the complex spatial patterns of ictal wavefront progression over a millimeter-scale cortical area in combination with spatially distributed measures of electrophysiology.

5. Line201-203 Ref36

It has been reported in Ref36 that a large area, high density electrode array can also be used to map electrographic seizures, and investigate the onset and propagation of epilepsy *in vivo*. Using the high density electrode array, detailed propagation spatial patterns has been recorded in ref 36. How does the multimodal recording using transparent microelectrodes here compared to the high density, large area electrophysiological method in ref36? To the reviewer's understanding, if using high-density electrode array, same results on seizure dynamics can be obtained even without the calcium imaging.

Even for the most advanced high-density surface μ ECoG electrode arrays currently available, such as the array reported in Ref. 36, the spatial resolution has been limited to approximately $500\mu\text{m}$.¹² This resolution is still orders of magnitudes larger than the spatial resolution achievable with calcium imaging methods. For the epifluorescence imaging used in this work, the resolution was $6.6\mu\text{m}$ per pixel, and significantly higher spatial resolution is achievable with today's most advanced imaging systems. The "effective" spatial resolution of μ ECoG may also be somewhat limited due to the unknown contributions of volume conduction and cell type to the observed signal. In contrast, calcium indicators can be genetically targeted to specific cell types such as subtypes of interneurons¹³ or astrocytes¹⁴, which offer a more localized, nuanced, and high-resolution picture of activity. The results regarding seizure dynamics that we present in this work could not be obtained from high-density electrophysiology alone, as the features from the calcium imaging, particularly those relating to the spatial extent of the ictal wavefront, have a much higher spatial resolution than could be obtained even with the most advanced surface μ ECoG arrays. We now include additional text in the discussion to help clarify the importance of combining both modalities in this work:

Line 341-348: Electrophysiological signals detected with μ ECoG are extracellular in origin, and this allows mixing of signals from different spatial origins and cell types through volume conduction. Thus, even with the most advanced high-density surface μ ECoG arrays, the effective spatial resolution is limited not only by the electrode spacing, but also by the contributions of volume conduction and the unknown cellular origins of the recorded signal. In contrast, calcium imaging relies on intracellular changes in calcium concentrations which can be more precisely localized to their cellular origins. Furthermore, calcium indicators can be genetically targeted to provide access to precisely isolated signals from specific neuronal subpopulations in a way that is not currently achievable by electrophysiology^{13,14}.

Reviewer #2:

In the manuscript, Driscoll and colleagues present simultaneous monitoring of epileptic seizures using electrophysiology recordings and fluorescence imaging in anesthetized 4-AP mouse model. This is a highly challenging experimental procedure and to achieve such data, the authors utilized graphene-based surface probes to create a semi-transparent electrophysiological probe that allows optical imaging through the probe. The main reasoning for such multi-modal measurements is to be able to acquire neural activity at temporal resolution of electrophysiology and spatial resolution of wide field imaging.

Overall, I found that the manuscript contained the necessary experimental data and explanations to support the authors' claims. However, there are a few points that I believe should be addressed to strengthen the manuscript.

1. The ECoG array used here is approximately (1.5 x 1.5 mm² , 16 electrodes) while the effective optical ROI is about (3 x 3 mm²) based on Figure 2d. Also, the dorsal surface of a mouse brain is 4x5 mm². Considering the capacity of the authors in fabricating high-density and high channel count probes, it would certainly be beneficial to discuss the advantages of the combinatory approach as opposed to increasing the spatial coverage of their minimally invasive electrophysiological probe to have both high spatial and temporal resolution with a single modality.

We thank the reviewer for raising this important point. Indeed, there is room to increase the spatial coverage of the electrodes presented here, and in certain contexts this may be preferable (e.g., where calcium imaging is not possible). However, the combined multimodal mapping approach in this study provides a means of truly localizing intracellular signal in way that electrophysiology alone cannot (because of volume conduction), as also described in response to Reviewer 1, point 5. To clarify this important point, we now include additional remarks on the importance of this combinatory approach in the main text, as follows:

Line 341-360: Electrophysiological signals detected with μ ECoG are extracellular in origin, and this allows mixing of signals from different spatial origins and cell types through volume conduction. Thus, even with the most advanced high-density surface μ ECoG arrays, the effective spatial resolution is limited not only by the electrode spacing, but also by the contributions of volume conduction and the unknown cellular origins of the recorded signal. In contrast, calcium imaging relies on intracellular changes in calcium concentrations which can be more precisely localized to their cellular origins. Furthermore, calcium indicators can be genetically targeted to provide access to precisely isolated signals from specific neuronal subpopulations in a way that is not currently achievable by electrophysiology^{13,14}.

We used widefield calcium epifluorescence imaging to offer a broad field of view encompassing the entire area covered by the 16-channel graphene array. Widefield imaging combined with somatically expressed calcium indicators is more sensitive to dendritic synaptic currents than individual action potentials, giving a readout of population activity rather than individual neuronal firing. For epileptic seizures, which emerge at the population level, this resolution is an appropriate spatial scale for subsequent analysis. However, some phenomena that characterize seizure activity (e.g. high frequency oscillations) cannot be captured with calcium-based methods at all, given the slow dynamics of both calcium transients and the fluorescent calcium indicators. Therefore, our approach allowed us to analyze the complex spatial patterns of ictal wavefront progression over a millimeter-scale cortical area in combination with spatially distributed measures of electrophysiology. In contrast to previous multimodal studies which have utilized cellular resolution multiphoton imaging, our approach also offers a much larger field of view.

2. Figure 2C: It would be beneficial to have multiple electrophysiological waveform traces to highlight the localized nature of the HFO.

Figure 2c has been updated to include wideband electrophysiology traces from 3 adjacent electrodes to highlight the localized nature of the HFOs. We have updated the figure and caption as follows:

Fig. 2: [...] c, Examples of two clinician-validated high frequency oscillations (HFOs) recorded on graphene electrodes. Raw, wideband signal from 3 adjacent electrodes highlights the localized nature of the HFOs. Below these, the 80-600 Hz bandpass-filtered signal and the spectrogram, which reveals 200-300 Hz power consistent with fast ripple HFOs, are shown for the bottom-most wideband signal trace.

3. Figure 2D (right): If I understood this graph correctly, the early time-points (blue) of the seizure have larger spatial extent than later time points (red). Is this a representative example?

In the seizure progression shown in Figure 2d (right), the seizure gradually expands and migrates to the right across the imaging field (light blue and yellow time points) before contracting at the later time points (red). Many seizures show a similar pattern where the spatial extent first expands from the seizure onset region and later contracts before the seizure ends. This example, along with the example shown in Figure 2d (left) are representative examples to illustrate the spatial complexity and diversity

seen in seizure spread dynamics and thus highlight the utility of using calcium imaging to capture these rich spatial dynamics. The word “representative” was added to the caption for Fig 2d to clarify this point.

4. Figure 3E: “we observe a correlation between normalized μ ECoG broadband power, and local average normalized calcium fluorescence Fig. 3e ” would benefit from further explanation and clarification. What is the correlation? Why is the heat map unit normalized frequency and not Hz?

Our apologies for the ambiguous labelling. The colorbar labelling refers to the frequency with which values on each bin were observed on the heat map, where the heat map effectively is a two-dimensional histogram. We have now labelled this ‘density’, with an additional explanation in the figure legend as follows:

Fig. 3: [...] e, Density plot indicating the significant relationship (Pearson’s correlation coefficient $r = 0.71$, $p < 0.001$ for the linear fit) between μ ECoG broadband power and local calcium fluorescence averages. The colormap indicates the density, or normalized observed frequency of values distributed along the two dimensions (μ ECoG broadband power, calcium fluorescence).

We have added the following to quantify the correlation: (Pearson’s correlation coefficient $r = 0.71$, $p < 0.001$ for linear fit) in both the **Figure 3 legend**, as shown above, and in **Line 247**.

5. A major pre-processing step of wide field imaging is dedicated toward extracting vasculature dilation. In this unique experiment, it would be interesting to provide comparison of neural activity patterns to changes in blood flow, which would be a direct proxy for fMRI studies. Although experimentally not necessary, the manuscript would certainly benefit from a discussion surrounding such opportunities in multimodal experiments.

This is exactly the sort of question we hope other groups may address in the future using the methods provided here. To facilitate this kind of work, we have made all of our data and code openly available for other scientists to utilize for their own questions. We have also included the following passage in the Discussion:

Line 478-483: Concurrently recorded multimodal data may be used to address novel biological questions that are not tractable with a single method approach. For example, genetically-targeted calcium imaging of inhibitory neuron activity during concurrent measurement of the μ ECoG may enable separating the contributions of excitatory and inhibitory cell types to the ECoG signal. Combining μ ECoG with imaging of cortical vasculature using injected fluorescent probes may also enable linking neural activity patterns to changes in blood flow, offering a direct proxy for fMRI studies¹⁵.

References:

1. Ding, D. *et al.* Evaluation of Durability of Transparent Graphene Electrodes Fabricated on Different Flexible Substrates for Chronic *in vivo* Experiments. *IEEE Trans. Biomed. Eng.* 1–1 (2020). doi:10.1109/TBME.2020.2979475
2. Wei, X. F. & Grill, W. M. Impedance characteristics of deep brain stimulation electrodes *in vitro* and *in vivo*. *J. Neural Eng.* **6**, 046008 (2009).

3. Sankar, V., Dieme, E. P. R., Sanchez, J. C., Prasad, A. & Nishida, T. Electrode impedance analysis of chronic tungsten microwire neural implants: Understanding abiotic vs. biotic contributions. *Front. Neuroeng.* **7**, 1–12 (2014).
4. Kuzum, D. *et al.* Transparent and flexible low noise graphene electrodes for simultaneous electrophysiology and neuroimaging. *Nat. Commun.* **5**, 5259 (2014).
5. Lempka, S. F. *et al.* Optimization of Microelectrode Design for Cortical Recording Based on Thermal Noise Considerations. *Annu. Int. Conf. IEEE Eng. Med. Biol. - Proc.* 3361–3364 (2006). doi:10.1109/IEMBS.2006.259432
6. Lempka, S. F. *et al.* Theoretical analysis of intracortical microelectrode recordings. *J. Neural Eng* **8**, 45006–45021 (2011).
7. D’Arsié, L. *et al.* Stable, efficient p-type doping of graphene by nitric acid. *RSC Adv.* **6**, 113185–113192 (2016).
8. Kasry, A., Kuroda, M. A., Martyna, G. J., Tulevski, G. S. & Bol, A. A. Chemical Doping of Large-Area Stacked Graphene Films for Use as Transparent, Conducting Electrodes. *ACS Nano* **4**, 3839–3844 (2010).
9. Bae, S. *et al.* Roll-to-roll production of 30-inch graphene films for transparent electrodes. *Nat. Nanotechnol.* **5**, 574–578 (2010).
10. Muldoon, S. F., Soltesz, I. & Cossart, R. Spatially clustered neuronal assemblies comprise the microstructure of synchrony in chronically epileptic networks. *Proc. Natl. Acad. Sci. U. S. A.* **110**, 3567–3572 (2013).
11. Stringer, C., Pachitariu, M., Steinmetz, N., Carandini, M. & Harris, K. D. High-dimensional geometry of population responses in visual cortex. *Nature* **571**, 361–365 (2019).
12. Viventi, J. *et al.* Flexible, foldable, actively multiplexed, high-density electrode array for mapping brain activity in vivo. *Nat. Neurosci.* **14**, 1599–605 (2011).
13. Aeed, F., Shnitzer, T., Talmon, R. & Schiller, Y. Layer- and Cell-Specific Recruitment Dynamics during Epileptic Seizures In Vivo. *Ann. Neurol.* **87**, 97–115 (2020).
14. Mariotti, L. *et al.* Interneuron-specific signaling evokes distinctive somatostatin-mediated responses in adult cortical astrocytes. *Nat. Commun.* **9**, 1–14 (2018).
15. Hirase, H., Creso, J. & Buzsáki, G. Capillary level imaging of local cerebral blood flow in bicuculline-induced epileptic foci. *Neuroscience* **128**, 209–216 (2004).

REVIEWERS' COMMENTS:

Reviewer #1 (Remarks to the Author):

The authors did a thorough job of addressing the issues brought up in the first review. Therefore, I would like to recommend acceptance of the manuscript for publication in Communications Biology.

Reviewer #2 (Remarks to the Author):

In this revised manuscript, authors substantially improved their manuscript from the previous version. I have no further comments and I am impressed by their research and manuscript.